# TRADE-OFF IN ESTIMATING THE NUMBER OF BYZANTINE CLIENTS IN FEDERATED LEARNING

## ABSTRACT

Federated learning has attracted increasing attention at recent large-scale optimization and machine learning research and applications, but is also vulnerable to Byzantine clients that can send any erroneous signals. Robust aggregators are commonly used to resist Byzantine clients. This usually requires to estimate the unknown number $f$ of Byzantine clients, and thus accordingly select the aggregators with proper degree of robustness (i.e., the maximum number $\hat{f}$ of Byzantine clients allowed by the aggregator). Such an estimation should have important effect on the performance, which has not been systematically studied to our knowledge. This work will fill in the gap by theoretically analyzing the worst-case error of aggregators as well as its induced federated learning algorithm for any cases of $\hat{f}$ and $f$. Specifically, we will show that underestimation ($\hat{f} < f$) can lead to arbitrarily poor performance for both aggregators and federated learning. For non-underestimation ($\hat{f} \geq f$), we have proved optimal lower and upper bounds of the same order on the errors of both aggregators and federated learning. All these optimal bounds are proportional to $\hat{f}/(n - f - \hat{f})$ with $n$ clients, which monotonically increases with larger $\hat{f}$. This indicates a fundamental trade-off: while an aggregator with a larger robustness degree $\hat{f}$ can solve federated learning problems of wider range $f \in [0, \hat{f}]$, the performance can deteriorate when there are actually fewer or even no Byzantine clients (i.e., $f \in [0, \hat{f})$).

## 1 INTRODUCTION

Federated learning proposed by (McMahan et al., 2017) is an important and popular framework for large-scale optimization and machine learning where multiple clients (i.e. computing devices) collaboratively optimize the objective function while keeping their local training data private. The most fundamental and common algorithm is federated averaging (FedAvg) (McMahan et al., 2017; Li et al., 2020; Collins et al., 2022), where the clients update their own model by using multiple steps of gradient-based approach on their own data, upload their updated models to the server, and download the average of these models from the server.

However, federated learning algorithms such as FedAvg is vulnerable to Byzantine clients (Lamport et al., 2019) which can upload arbitrary model to the server. A fundamental and popular way to make federated learning algorithm robust to Byzantine clients is to apply a robust aggregation to the uploaded models to filter outliers (Li et al., 2021a; 2023; Allouah et al., 2024). Some representative aggregations include geometric median (GM) (Chen et al., 2017; Pillutla et al., 2022), coordinate-wise trimmed mean (CWTM) (Yin et al., 2018), coordinate-wise median (CWMed) (Yin et al., 2018), Krum (Blanchard et al., 2017), centered clipping (Karimireddy et al., 2021), clustering (Sattler et al., 2020; Li et al., 2021b), etc., as summarized and empirically compared in (Li et al., 2023). To select an aggregator with a proper breakpoint $\hat{f}$ (namely, the number of maximum tolerable Byzantine clients) is essential to ensure good federated learning performance (Karimireddy et al., 2022; Gupta et al., 2023; Allouah et al., 2024; Otsuka et al., 2025). However, there lacks a systematic study on the effect of the selected $\hat{f}$ when there are actually $f$ Byzantine clients. Therefore, it is natural to ask the following fundamental research question:

> **Q:** *Applying an algorithm that can resist $\hat{f}$ Byzantine clients to federated learning problem with $f$ actual Byzantine clients, what is the effect of the estimated number $\hat{f}$ on the performance?*

## 1.1 Our Contributions

To our knowledge, this work for the first time systematically investigates the theoretical effect of the estimated number of Byzantine clients on federated learning performance. In particular, we theoretically prove that underestimation of $f$ (i.e. $\hat{f} < f$) can lead to arbitrarily poor performance on both aggregation error and convergence of the commonly used Federated Robust Averaging (FedRo) algorithm, even if the objective function satisfies the Polyak-Łojasiewicz (PŁ) condition that is amenable to global convergence. When $\hat{f} \geq f$ (non-underestimation), we obtain the lower and upper bounds of the aggregation error and convergence rate of FedRo. Each lower bound is tight as its order matches the corresponding upper bound. All these tight bounds are proportional to $\frac{\hat{f}}{n-f-\hat{f}}$ with $n$ clients, which is monotonically increasing as $\hat{f}$ increases from $f$. This indicates a fundamental trade-off: while an aggregator with a larger robustness degree $\hat{f}$ can solve federated learning problems of wider range $f \in [0, \hat{f}]$, the performance can deteriorate when there are actually fewer or even no Byzantine clients (i.e., $f \in [0, \hat{f})$).

## 1.2 Related Works

Some federated learning works (Bagdasaryan et al., 2020; Tolpegin et al., 2020; Wang et al., 2020a; Xie et al., 2020) focus on targeted attacks (i.e., back-door attacks) that fool the global model to predict certain samples with some incorrect targeted labels, while this work focuses on untargeted attacks (i.e., Byzantine attacks) (Li et al., 2023; Allouah et al., 2024; Xu et al., 2025) that hamper the overall learning performance with no specific focus. In addition to aggregation-based approach of our focus, various other approaches have been proposed to resist Byzantine clients in federated learning. Xie et al. (2019); Cao et al. (2021); Park et al. (2021); Kritharakis et al. (2025) allow the server to preserve some representative data samples to evaluate and select the uploaded models, which is not always possible in practice since these representative samples can be similar to those on the local clients and thus raise privacy concern (Xu et al., 2025). Panda et al. (2022); Meng et al. (2023); Zhang & Hu (2023); Xu et al. (2025) sparsify the model updates to alleviate the effect of Byzantine clients.

## 2 Preliminaries

**Federated Learning Problem with Byzantine Clients:** In standard federated learning, there is a server communicating with $n$ clients. Among the $n$ clients indexed by $[n] \stackrel{\text{def}}{=} \{1, 2, \ldots, n\}$ respectively, there are $f$ Byzantine clients sending any erroneous signals (Byzantine attack) to interfere with the server. Denote $\mathcal{H}$ as the set of the other honest clients, with size $|\mathcal{H}| = n - f$. The server can only communicate model parameters with the clients, but does not know the number and identity of the Byzantine clients. These honest clients aim to collaboratively solve the following optimization problem, under the interference of the $f$ Byzantine clients.

$$\min_{w \in \mathbb{R}^d} \left\{ \ell_{\mathcal{H}}(w) \stackrel{\text{def}}{=} \frac{1}{|\mathcal{H}|} \sum_{k \in \mathcal{H}} \ell_k(w) \right\}. \tag{1}$$

where the loss function $\ell_k : \mathbb{R}^d \to \mathbb{R}$ is associated with the local private data in the $k$-th client, and is thus unknown to the server and the other clients. Here, we assume that less than half of the clients are Byzantine (i.e., $f < \frac{n}{2}$) since this problem has been proved intractable otherwise (Liu et al., 2021).

**Federated Robust Averaging (FedRo) Algorithm:** The Federated Robust Averaging (FedRo) algorithm (as shown in Algorithm 1) is commonly used to solve the federated learning problem (1) with Byzantine clients (Li et al., 2021a; 2023; Allouah et al., 2024). In each communication round, every client downloads the model $w_t$ from the server. Then every honest client $k \in \mathcal{H}$ performs local gradient descent updates (2) $H$ times on its local loss function $\ell_k$, while the Byzantine clients can upload arbitrary vectors to interfere with the learning. At the end of each round, the

server updates its global parameter by aggregating the uploaded vectors. Since the server does not know which clients are Byzantine, the selection of the aggregator $\mathcal{A} : (\mathbb{R}^d)^n \to \mathbb{R}^d$ is essential to ensure the updated global model is robust to the Byzantine attacks. When selecting the averaging aggregator $\mathcal{A}(\{x_k\}_{k=1}^n) = \frac{1}{n}\sum_{k=1}^n x_k$, the FedRo algorithm reduces to the popular federated averaging (FedAvg) algorithm (McMahan et al., 2017; Li et al., 2020; Collins et al., 2022) which is vulnerable to Byzantine clients.

---

**Algorithm 1** FedRo: Federated Robust Averaging Algorithm

---

**Input:** The set of honest clients $\mathcal{H} \subset [n]$ with size $|\mathcal{H}| = n - f$ ($0 \leq f < \frac{n}{2}$), number of rounds $T$, number of local parameter updates $H$, initial parameter $w_0 \in \mathbb{R}^d$, stepsize $\gamma_t$, aggregator $\mathcal{A}$.

**for** communication rounds $t = 0, 1, \ldots, T - 1$ **do**

    The server sends $w_t$ to every clients.

    **for** all clients $k \in [n]$ **in parallel do**

        **if** $k \in \mathcal{H}$ (honest client) **then**

            Initialize $w_{t,0}^{(k)} = w_t$.

            **for** $h = 0, \ldots, H - 1$ **do**

                Local parameter update:

$$w_{t,h+1}^{(k)} = w_{t,h}^{(k)} - \gamma_t \nabla \ell_k(w_{t,h}^{(k)}) \tag{2}$$

            **end**

            Upload $w_t^{(k)} = w_{t,H}^{(k)}$ to the server.

        **else**

            The Byzantine client $k$ uploads arbitrary $w_t^{(k)} \in \mathbb{R}^d$ to the server.

        **end**

    **end**

    The server updates the global parameter using aggregator $\mathcal{A}$.

$$w_{t+1} = w_t + \mathcal{A}(\{w_t^{(k)} - w_t\}_{k=1}^n). \tag{3}$$

**end**

**Output:** Select a parameter from $\{w_t\}_{t=0}^{T-1}$ uniformly at random.

---

## 3 AGGREGATION ERROR ANALYSIS

The aggregator $\mathcal{A}$ is the core of the robust federated learning algorithms such as FedRo (Algorithm 1), so it is essential to select a proper aggregator with certain robustness properties to Byzantine clients. Multiple robustness metrics have been proposed. This work will focus on the following $(f, \kappa)$-robustness (Allouah et al., 2023) which unifies the other robustness metrics including $(f, \lambda)$-resilient averaging (Farhadkhani et al., 2022) and $(\delta_{\max}, c)$-ARAgg (Karimireddy et al., 2022).

**Definition 1** ($(f, \kappa)$-robust aggregator). *For any $\kappa \geq 0$ and integer $0 \leq f < \frac{n}{2}$, an aggregator $\mathcal{A} : (\mathbb{R}^d)^n \to \mathbb{R}^d$ is called $(f, \kappa)$-robust if for any $x_1, \ldots, x_n \in \mathbb{R}^d$ and any $S \subset [n] \overset{\text{def}}{=} \{1, 2, \ldots, n\}$ with size $|S| = n - f$, we have*

$$\|\mathcal{A}(\{x_k\}_{k=1}^n) - \overline{x}_S\|^2 \leq \frac{\kappa}{|S|} \sum_{i \in S} \|x_i - \overline{x}_S\|^2 \tag{4}$$

*where $\overline{x}_S = \frac{1}{|S|} \sum_{i \in S} x_i$, and $\|\mathcal{A}(\{x_k\}_{k=1}^n) - \overline{x}_S\|^2$ is called the aggregation error.*

Since the server does not know which clients are Byzantine, the aggregation error bound (4) should hold for any set $S$ after removing $f$ possibly Byzantine clients. Therefore, an $(f, \kappa)$-robust aggregator can resist $f$ Byzantine clients with robustness coefficient $\kappa$. Some commonly used aggregators have been proved $(f, \kappa)$-robust, such as geometric median (GM), coordinate-wise trimmed mean (CWTM), coordinate-wise median (CWMed), Krum, etc. (Allouah et al., 2023), as shown in the first row of Table 1. Ideally, one can apply an $(f, \kappa)$-robust aggregator when there are actually $f$ Byzantine

clients. However, $f$ is usually unknown in practice, so we can only use its estimation $\hat{f}$ and apply an $(\hat{f}, \hat{\kappa})$ ($\hat{\kappa} > 0$) aggregator. This section will analyze the aggregation error of an $(\hat{f}, \hat{\kappa})$-robust aggregator when there are actually $f$ Byzantine clients, for both underestimation ($\hat{f} < f$) and non-underestimation ($\hat{f} \geq f$). To our knowledge, the existing literature has only studied a small number of special cases, including $f = \hat{f}$ (exact estimation) (Allouah et al., 2023) and $f = 0$ (no Byzantine clients) (Yang et al., 2025) as shown in the first two rows of Table 1. Throughout this work, we always assume that $\hat{f}, f \in \left[0, \frac{n}{2}\right)$ to ensure tractability.

Table 1: The value of $\kappa$ for $(\hat{f}, \hat{\kappa})$-robust aggregator that is also $(f, \kappa)$-robust. We use the names $\hat{f}$-Krum, $\hat{f}$-NNM and $TM_{\hat{f}/n}$ to stress their dependence on the parameter $\hat{f}$ for clarity, while geometric median (GM) and coordinate-wise median (CWMed) do not depend on $\hat{f}$. The composite aggregator $\hat{f}$-Krum$\circ\hat{f}$-NNM is defined by Definition 3 with $\mathcal{A} = \hat{f}$-Krum.

| $f$ | GM | CWTM $TM_{\hat{f}/n}$ | CWMed | $\hat{f}$-Krum | $\hat{f}$-Krum$\circ\hat{f}$-NNM | **Lower bound** |
|---|---|---|---|---|---|---|
| $f = \hat{f}$ (Allouah et al., 2023) | $4\left(\frac{n-\hat{f}}{n-2\hat{f}}\right)^2$ | $\frac{6\hat{f}}{n-2\hat{f}}\left(\frac{n-\hat{f}}{n-2\hat{f}}\right)$ | $4\left(\frac{n-\hat{f}}{n-2\hat{f}}\right)^2$ | $6\left(\frac{n-\hat{f}}{n-2\hat{f}}\right)$ | - | $\frac{\hat{f}}{n-2\hat{f}}$ |
| $f = 0$ (Yang et al., 2025) | $1$ | $\frac{\hat{f}}{n-\hat{f}}$ | $\frac{\left\lfloor\frac{n-1}{2}\right\rfloor}{n-\left\lfloor\frac{n-1}{2}\right\rfloor}$ | - | - | $\frac{\hat{f}}{n-\hat{f}}$ |
| $f \leq \hat{f}$ (Theorem 2) | - | - | - | - | $\frac{84\hat{f}}{n-f-\hat{f}}$ | $\frac{\hat{f}}{n-f-\hat{f}}$ |

## 3.1 Aggregation Error for Underestimation ($\hat{f} < f$)

**Theorem 1.** *For any $\hat{f} \in \left(0, \frac{n}{2}\right)$ and $\hat{\kappa} > 0$, there exists an $(\hat{f}, \hat{\kappa})$-robust aggregator that is not $(f, \kappa)$-robust for any $f \in \left(\hat{f}, \frac{n}{2}\right)$ and $\kappa > 0$.*

**Remark:** Theorem 1 indicates that an $(\hat{f}, \hat{\kappa})$-robust aggregator does not necessarily tolerate more than $\hat{f}$ Byzantine clients. Therefore, if we underestimate the number $f$ of Byzantine clients, the aggregator can have arbitrarily inaccurate performance.

A typical example of such an aggregator satisfying Theorem 1 is the commonly used coordinate-wise trimmed mean (CWTM) aggregator (Yin et al., 2018; Allouah et al., 2023; Yang et al., 2025), as defined below.

**Definition 2.** *For any $x_1, \ldots, x_n \in \mathbb{R}^d$, the CWTM aggregator denoted as $TM_{\hat{f}/n} : (\mathbb{R}^d)^n \to \mathbb{R}^d$ is defined as follows*

$$[TM_{\hat{f}/n}(x_1, \ldots, x_n)]_j = \frac{1}{n - 2\hat{f}} \sum_{x \in \mathcal{X}_j} x, \tag{5}$$

*where $[v]_j$ denotes the $j$-th coordinate of a vector $v$, and the set $\mathcal{X}_j$ is obtained by deleting the $\hat{f}$ smallest and $\hat{f}$ largest values from $\{[x_i]_j\}_{i=1}^n$.*

In other words, $TM_{\hat{f}/n}$ averages the remaining $n - 2\hat{f}$ samples of each coordinate after removing the $\hat{f}$ largest and $\hat{f}$ smallest samples. $TM_{\hat{f}/n}$ has been proved to be an $(\hat{f}, \hat{\kappa})$-robust aggregator (see Proposition 2 of Allouah et al. (2023)) with $\hat{\kappa} = \frac{6\hat{f}}{n-2\hat{f}}\left(1 + \frac{\hat{f}}{n-2\hat{f}}\right)$. However, if there are $f$ ($f > \hat{f}$) extremely large (or small) $x_i$ given by Byzantine clients, then after removing $\hat{f}$ largest (smallest) elements, the remaining extreme values can still heavily affect the average (5). This intuition motivates the counter example for proving Theorem 1 in Appendix A.

## 3.2 Aggregation Error for Non-underestimation ($\hat{f} \geq f$)

When the estimated number $\hat{f}$ of Byzantine clients is not less than the true number $f$, any $(\hat{f}, \hat{\kappa})$-robust aggregator satisfies the following important properties.

**Theorem 2.** *For any $0 \leq f \leq \hat{f} < \frac{n}{2}$, an $(\hat{f}, \hat{\kappa})$-robust aggregator $\mathcal{A}$ satisfies:*

1. $\mathcal{A}$ is always $(f, \kappa)$-robust with $\kappa = \hat{\kappa}$.

2. If $\mathcal{A}$ is $(f, \kappa)$-robust, then $\kappa \geq \frac{\hat{f}}{n-f-\hat{f}}$.

3. Furthermore, there exists such an $\mathcal{A}$ that is $(f, \kappa)$-robust with $\kappa \leq \frac{84\hat{f}}{n-f-\hat{f}}$.

**Remark:** In Theorem 2, Item 1 shows that an $(\hat{f}, \hat{\kappa})$-robust aggregator is able to tackle any problem with fewer Byzantine clients $f \leq \hat{f}$. However, the robust coefficient $\kappa$ has a lower bound $\frac{\hat{f}}{n-f-\hat{f}}$ by Item 2, which reduces to the existing lower bound $\frac{\hat{f}}{n-2\hat{f}}$ when $f = \hat{f}$ (Allouah et al., 2023) and to $\frac{\hat{f}}{n-\hat{f}}$ when $f = 0$ (Yang et al., 2025). This lower bound $\frac{\hat{f}}{n-f-\hat{f}}$ is order-optimal since a certain $(\hat{f}, \hat{\kappa})$-robust aggregator can achieve the order-matching upper bound $\kappa \leq \mathcal{O}\left(\frac{\hat{f}}{n-f-\hat{f}}\right)$ by Item 3. Note that as $\hat{f}$ increases from $f$, this order-optimal bound $\mathcal{O}\left(\frac{\hat{f}}{n-f-\hat{f}}\right)$ is increasing, which yields a larger aggregation error.

### 3.3 PROOF SKETCH FOR ITEM 3 OF THEOREM 2

Item 3 is the most challenging to prove in Theorem 2. One challenge is to find out such a proper aggregator with order-matching lower bound $\kappa \leq \mathcal{O}\left(\frac{\hat{f}}{n-f-\hat{f}}\right)$ under $f$ Byzantine clients, partially since $\kappa$ of the commonly used aggregators like GM, CWTM, CWMed and Krum do not match the lower bound even in the simple special case of $f = \hat{f}$, as shown in the first row of Table 1 [1]. To improve $\kappa$ of these aggregators, we will composite them with the nearest neighbor mixing (NNM) proposed by (Allouah et al., 2023), an aggregator booster defined as follows.

**Definition 3.** *For any $\hat{f} \in \left[0, \frac{n}{2}\right)$, $k \in [n]$ and $x_1, \ldots, x_n \in \mathbb{R}^d$, denote $\mathcal{N}_k \subset [n]$ as the set of $(n - \hat{f})$ indexes from $[n]$ such that $\{x_i\}_{i \in \mathcal{N}_k}$ are the $(n - \hat{f})$ nearest neighbors of $x_k$. In other words, $\max_{i \in \mathcal{N}_k} \|x_i - x_k\| \leq \min_{j \in [n] \setminus \mathcal{N}_k} \|x_j - x_k\|$. The mapping $\hat{f}$-NNM: $(\mathbb{R}^d)^n \to (\mathbb{R}^d)^n$ is defined as follows.*

$$\hat{f}\text{-NNM}(x_1, \ldots, x_n) = (y_1, \ldots, y_n), \text{ where } y_k = \frac{1}{n - \hat{f}} \sum_{i \in \mathcal{N}_k} x_i. \tag{6}$$

*For any aggregator $\mathcal{A} : (\mathbb{R}^d)^n \to \mathbb{R}^d$, define the composite aggregator $(\mathcal{A} \circ \hat{f}\text{-NNM})(x_1, \ldots, x_n) = \mathcal{A}(y_1, \ldots, y_n)$, with the notations in Eq. (6).*

Lemma 1 of (Allouah et al., 2023) has proved that for any $(\hat{f}, \hat{\kappa})$-robust aggregator $\mathcal{A}$, the composite aggregator $\mathcal{A} \circ \hat{f}$-NNM is $(\hat{f}, \kappa')$-robust with improved robustness coefficient $\kappa' \leq \frac{8\hat{f}(\hat{\kappa}+1)}{n-\hat{f}}$. Selecting $\mathcal{A} = \hat{f}$-Krum which is $(\hat{f}, \hat{\kappa})$-robust with $\hat{\kappa} = 6\left(\frac{n-\hat{f}}{n-2\hat{f}}\right)$, the composite aggregator has $\kappa' \leq \frac{56\hat{f}}{n-2\hat{f}}$, which matches the lower bound $\frac{\hat{f}}{n-2\hat{f}}$ when $f = \hat{f}$ (see the first row of Table 1). Therefore, it is natural to consider boosting $\hat{f}$-Krum with NNM. We extend the boosting property of NNM to the case of $f \leq \hat{f}$ as follows, which preserves the order of $\kappa' \leq \frac{8\hat{f}(\hat{\kappa}+1)}{n-\hat{f}}$ in (Allouah et al., 2023) when $f = \hat{f}$.

**Lemma 1.** *For any $0 \leq f \leq \hat{f} < \frac{n}{2}$ and any $(f, \kappa)$-robust aggregator $\mathcal{A}$, the composition $\mathcal{A} \circ (\hat{f}\text{-NNM})$ is an $(f, \kappa')$-robust aggregator with $\kappa' \leq \frac{12\hat{f}(\kappa+1)}{n-f}$.*

We will prove Lemma 1 in Appendix B.4.

---

[1] CWTM has $\kappa = \frac{6f}{n-2f}\left(1 + \frac{f}{n-2f}\right)$ which has higher order than the lower bound $\frac{f}{n-2f}$ if $n - 2f \ll n$. Krum, GM and CWMed have $\kappa = 6\left(1 + \frac{f}{n-2f}\right)$, $\kappa = 4\left(1 + \frac{f}{n-2f}\right)^2$ and $\kappa = 4\left(1 + \frac{f}{n-2f}\right)^2$ respectively. All of them have higher order than the lower bound $\frac{f}{n-2f}$ if $f \ll n$.

Then we investigate the following $\hat{f}$-Krum aggregator (Blanchard et al., 2017; Allouah et al., 2023; Yang et al., 2025).

**Definition 4.** *The $\hat{f}$-Krum aggregator is defined as the following mapping $(\mathbb{R}^d)^n \to \mathbb{R}^d$.*

$$(\hat{f}\text{-Krum})(x_1, \ldots, x_n) = x_{k^*}, \text{where } k^* = \arg\min_{k \in [n]} \sum_{i \in \mathcal{N}_k} \|x_i - x_k\|, \tag{7}$$

*where $\mathcal{N}_k \subset [n]$ is the set of $(n - \hat{f})$ indexes from $[n]$ such that $\{x_i\}_{i \in \mathcal{N}_k}$ are the $(n - \hat{f})$ nearest neighbors of $x_k$. In other words, $\max_{i \in \mathcal{N}_k} \|x_i - x_k\| \leq \min_{j \in [n] \setminus \mathcal{N}_k} \|x_j - x_k\|$.*

In Appendix B.5, we will prove the following property of $\hat{f}$-Krum aggregator when applying to the scenario with $f$ Byzantine clients, which exactly reduces to $\kappa = \frac{6(n-\hat{f})}{n-2\hat{f}}$ obtained in Proposition 3 of (Allouah et al., 2023) when $f = \hat{f}$.

**Lemma 2.** *For any $0 \leq f \leq \hat{f} < \frac{n}{2}$, $\hat{f}$-Krum is an $(f, \kappa)$-robust aggregator with $\kappa = \frac{6(n-f)}{n-f-\hat{f}}$.*

Based on Lemmas 1 and 2, for any $0 \leq f \leq \hat{f} < \frac{n}{2}$, $(\hat{f}\text{-Krum}) \circ (\hat{f}\text{-NNM})$ is an $(f, \kappa)$-robust aggregator with

$$\kappa = \frac{12\hat{f}}{n-f}\Big(\frac{6(n-f)}{n-f-\hat{f}} + 1\Big) \leq \frac{12\hat{f}}{n-f} \cdot \Big(\frac{6(n-f)}{n-f-\hat{f}} + \frac{n-f}{n-f-\hat{f}}\Big) = \frac{84\hat{f}}{n-f-\hat{f}}.$$

This concludes the proof.

### 3.4 SUMMARY: TRADE-OFF IN ESTIMATING $f$ FOR AGGREGATORS

We have analyzed the aggregation error in this section when applying $(\hat{f}, \hat{\kappa})$-robust aggregator to a setting with $f$ Byzantine clients. Theorem 1 shows we should by no means underestimate the number of Byzantine clients (i.e. $\hat{f} < f$) since that can lead to arbitrarily poor performance. While Theorem 2 shows that non-underestimation ($\hat{f} \geq f$) can tackle this problem, the order-optimal lower bound of robustness coefficient $\kappa \geq \frac{\hat{f}}{n-f-\hat{f}}$ increases (i.e., increased aggregation error) as $\hat{f}$ increases. Therefore, it is recommended to reduce the overestimation amount $\hat{f} - f \geq 0$ as much as possible, and the exact estimation $\hat{f} = f$ yields the optimal performance. This indicates a fundamental trade-off in estimating $f$, while a highly robust aggregator with larger $\hat{f}$ can tackle a wider range of settings for any $f \leq \hat{f}$, the aggregation error is also larger for any fixed $f$. The next section will prove a similar order-optimal bound and trade-off on federated learning.

## 4 CONVERGENCE ANALYSIS

To analyze the convergence of Algorithm 1, we adopt the following standard assumptions on the federated optimization problem (1) below.

**Assumption 1** (Loss bound). *The objective function (1) admits a finite minimum value denoted as $\ell^* \overset{\text{def}}{=} \inf_{w \in \mathbb{R}^d} \ell_{\mathcal{H}}(w) \in \mathbb{R}$.*

**Assumption 2** (Smoothness). *Each individual function $\ell_k$ is $L$-smooth for some $L > 0$, that is, for any $w, w' \in \mathbb{R}^d$, we have $\|\nabla \ell_k(w') - \nabla \ell_k(w)\| \leq L\|w' - w\|$.*

**Assumption 3** (Heterogeneity bound). *There exists a constant $G > 0$ such that*

$$\frac{1}{|\mathcal{H}|} \sum_{k \in \mathcal{H}} \|\nabla \ell_k(w) - \nabla \ell_{\mathcal{H}}(w)\|^2 \leq G^2. \tag{8}$$

**Assumption 4** (Polyak-Łojasiewicz (PŁ) condition). *There exists a constant $\mu > 0$ such that $\ell_{\mathcal{H}}$ is $\mu$-PŁ gradient dominant, that is, $\ell_{\mathcal{H}}(w) - \ell^* \leq \frac{1}{2\mu}\|\nabla \ell_{\mathcal{H}}(w)\|^2$ for any $w \in \mathbb{R}^d$.*

Assumptions 1-3 are popular in distributed and federated learning (Allouah et al., 2023; Errami & Bergou, 2024; Allouah et al., 2024; Yang et al., 2025; Otsuka et al., 2025). In particular, a larger

$G^2$ in Assumption 3 means the honest clients have more heterogeneous data, which makes the federated optimization problem (1) more challenging. The notion of PŁcondition (Assumption 4) proposed by (Polyak, 1963) is widely used in nonconvex optimization to guarantee global convergence (Karimi et al., 2016; Chakrabarti & Baranwal, 2024; Yang et al., 2025). With these assumptions, we will analyze the convergence rate of Algorithm 1 with an $(\hat{f}, \hat{\kappa})$-robust aggregator on federated learning with $f$ Byzantine clients. We will discuss in two cases, underestimation ($\hat{f} < f$) and non-underestimation ($\hat{f} \geq f$).

## 4.1 DIVERGENCE FOR UNDERESTIMATION ($\hat{f} < f$)

**Theorem 3.** *For any $0 \leq \hat{f} < f < \frac{n}{2}$, there exist an $(\hat{f}, \hat{\kappa})$-robust aggregator $\mathcal{A}$, a set $\mathcal{H} \subset [n]$ with size $|\mathcal{H}| = n - f$, Byzantine clients' strategies and loss functions $\{\ell_i\}_{i \in \mathcal{H}}$ satisfying Assumptions 1-4 such that when implementing Algorithm 1 with the aggregator $\mathcal{A}$, any initialization $w_0$ and constant stepsize $\gamma_t = \gamma > 0$, the generated sequence $w_t$ diverges as follows as $T \to +\infty$.*

$$\frac{1}{T} \sum_{t=0}^{T-1} \|\nabla \ell_{\mathcal{H}}(w_t)\|^2 \to +\infty \tag{9}$$

$$\ell_{\mathcal{H}}(w_T) - \ell^* \to +\infty \tag{10}$$

**Remark:** Theorem 3 indicates that if the aggregator is only robust to $\hat{f}$ Byzantine clients that is fewer than the actual number $f$ of Byzantine clients, then Algorithm 1 with any initialization and stepsize can diverge in some federated learning problems. Therefore, we should always guarantee the non-underestimation condition that $\hat{f} \geq f$.

## 4.2 CONVERGENCE RATE FOR NON-UNDERESTIMATION ($\hat{f} \geq f$)

**Theorem 4** (Lower bound). *For any $0 \leq f \leq \hat{f} < \frac{n}{2}$ and any $(\hat{f}, \hat{\kappa})$-robust aggregator $\mathcal{A}$, there exist $\mathcal{H} \subset [n]$ with size $|\mathcal{H}| = n - f$, Byzantine clients' strategies and loss functions $\{\ell_i\}_{i \in \mathcal{H}}$ satisfying Assumptions 1-4 such that for any initialization $w_0$ and constant stepsize $\gamma_t = \gamma > 0$, the sequence $w_t$ generated from Algorithm 1 with aggregator $\mathcal{A}$ either does not change over iteration (i.e., $w_t \equiv w_0$) or satisfies the following convergence lower bounds.*

$$\limsup_{T \to \infty} \frac{1}{T} \sum_{t=0}^{T-1} \|\nabla \ell_{\mathcal{H}}(w_t)\|^2 \geq \frac{\hat{f} G^2}{n - f - \hat{f}} \tag{11}$$

$$\limsup_{T \to \infty} \ell_{\mathcal{H}}(w_T) - \ell^* \geq \frac{\hat{f} G^2}{2\mu(n - f - \hat{f})} \tag{12}$$

**Remark:** In most cases, the heterogeneity $G^2 > 0$. Then regardless of the hyperparameter choices, Algorithm 1 with an $(\hat{f}, \hat{\kappa})$-robust aggregator cannot converge to a stationary or optimal point of some objective functions, since the convergence metric of gradients and function value are lower bounded by $\frac{\hat{f} G^2}{n - f - \hat{f}} > 0$ and $\frac{\hat{f} G^2}{2\mu(n - f - \hat{f})} > 0$ respectively as shown above. These lower bounds increase as $\hat{f}$ increases from $f$. Later, we will show that these lower bounds are tight since their orders match the upper bounds in the upcoming Theorem 5.

**Proof Sketch of Theorem 4:** Select the following loss functions with scalar input $w \in \mathbb{R}$, which can be verified to satisfy Assumptions 1-4.

$$\ell_k(w) = \begin{cases} cG(w+1)^2, k = 1, 2, \ldots, \hat{f} \\ cGw^2, k = \hat{f} + 1, \hat{f} + 2, \ldots, n \end{cases}, \tag{13}$$

where $c = \frac{n - f}{2\sqrt{\hat{f}(n - f - \hat{f})}}$. Suppose $\mathcal{H} = [n - f] = \{1, 2, \ldots, n - f\}$ and the Byzantine clients adopt the same honest behavior as the honest clients, i.e., upload the result after $H$ local gradient descent updates (2). Then the global model updates as follows.

$$w_{t+1} = \Gamma^H w_t, \text{ where } \Gamma = 1 - 2cG\gamma.$$

Then we can prove Theorem 4 in four cases of the hyperparameters $\gamma$ and $H$ that respectively satisfy $|\Gamma^H| < 1$, $\Gamma^H = 1$, $\Gamma^H = -1$ and $|\Gamma^H| > 1$. See the whole proof in Appendix D.

**Theorem 5** (Upper bound). *Suppose Assumptions 1-3 hold. Apply Algorithm 1 with an $(f, \kappa)$-aggregator and stepsize $\gamma = \frac{1}{c'LHT^{1/3}}$ ($c' = \max(4\sqrt{2}, \sqrt{384\kappa})$) to the federated optimization problem (1) with $f$ Byzantine clients. The algorithm output $\{w_t\}_{t=0}^{T-1}$ satisfies the following convergence rate.*

$$\frac{1}{T}\sum_{t=0}^{T-1}\|\nabla\ell_{\mathcal{H}}(w_t)\|^2 \leq \frac{16c'LH[\ell_{\mathcal{H}}(w_0) - \ell^*] + G^2}{T^{2/3}} + 90\kappa G^2. \tag{14}$$

*Furthermore, under Assumption 4 (i.e., $\ell_{\mathcal{H}}$ satisfies the PŁ condition), we can select stepsize $\gamma = \frac{1}{c'LHT^{1-\beta}}$ for any $\beta \in (0, 1)$ which yields the following convergence rate.*

$$\ell_{\mathcal{H}}(w_T) - \ell^* \leq \exp\Big(-\frac{\mu T^\beta}{8c'L}\Big)[\ell_{\mathcal{H}}(w_0) - \ell^*] + \frac{G^2}{2\mu T^{2-2\beta}} + \frac{45\kappa G^2}{\mu}. \tag{15}$$

**Remark:** As $T \to +\infty$, the upper bounds above respectively converge to $90\kappa G^2$ and $\frac{45\kappa G^2}{\mu}$. Moreover, if Algorithm 1 uses an $(\hat{f}, \hat{\kappa})$-robust aggregator with $\hat{f} \geq f$, which is also $(f, \kappa)$-robust that can achieve the order-optimal upper bound $\kappa \leq \mathcal{O}\big(\frac{\hat{f}}{n-f-\hat{f}}\big)$ by Theorem 2, then the upper bounds (14) and (15) respectively converge to $\mathcal{O}\big(\frac{\hat{f}G^2}{n-f-\hat{f}}\big)$ and $\mathcal{O}\big(\frac{\hat{f}G^2}{\mu(n-f-\hat{f})}\big)$, which are tight since they match the orders of the lower bounds in Theorem 4.

## 4.3 SUMMARY: TRADE-OFF IN ESTIMATING $f$ FOR FEDERATED LEARNING

We have analyzed the convergence of Algorithm 1 with an $(\hat{f}, \hat{\kappa})$-robust aggregator on federated learning with $f$ Byzantine clients. Theorem 3 indicates that we should always avoid underestimation ($\hat{f} < f$) as that can lead to divergence. When $\hat{f} \geq f$, Theorem 4 provides convergence lower bounds that match the orders of the upper bounds in Theorem 5. These results are analogous to those for the robustness coefficient $\kappa$ of the aggregator analyzed in Section 3. We summarize all these main theoretical results in Table 2, which shows that the order-optimal bounds for both $\kappa$ and the two federated learning convergence metrics are proportional to $\frac{\hat{f}}{n-f-\hat{f}}$ which increases as $\hat{f}$ increases from $f$. Therefore, there is a fundamental trade-off in estimating $f$: Algorithm 1 with aggregators robust to more Byzantine clients has degraded performance in terms of both aggregation error and algorithm convergence, when there are actually not that many Byzantine clients.

Table 2: Summary of our main theoretical results. As we apply an $(\hat{f}, \hat{\kappa})$-robust aggregator to federated learning with $f$ Byzantine clients, we show three performance metrics: robust coefficient $\kappa$ of the aggregator (also $(f, \kappa)$-robust) and the two convergence metrics for Algorithm 1, under Assumptions 1-4. These metrics can be arbitrarily poor when $\hat{f} < f$, and have the following lower bounds that match the order of the corresponding upper bounds when $\hat{f} \geq f$.

| | $\kappa$ | $\frac{1}{T}\sum_{t=0}^{T-1}\|\nabla\ell_{\mathcal{H}}(w_t)\|^2$ | $\ell_{\mathcal{H}}(w_T) - \ell^*$ |
|---|---|---|---|
| Underestimation | Possibly non-exist | Possibly $\to +\infty$ | Possibly $\to +\infty$ |
| ($\hat{f} < f$) | (Theorem 1) | (Theorem 3) | (Theorem 3) |
| Non-underestimation | $\mathcal{O}\big(\frac{\hat{f}}{n-f-\hat{f}}\big)$ | $\mathcal{O}\big(\frac{\hat{f}G^2}{n-f-\hat{f}}\big)$ | $\mathcal{O}\big(\frac{\hat{f}G^2}{\mu(n-f-\hat{f})}\big)$ |
| ($\hat{f} \geq f$) | (Theorem 2) | (Theorems 4-5) | (Theorems 4-5) |

**Comparison with Related Works:** Two recent works have obtained results that are similar to part of our results. Allouah et al. (2024) obtains a near-optimal convergence rate of Byzantine-robust federated learning algorithm where a random subset of $\hat{n}$ clients participate in each communication round, which relies on the effect of the estimated number $\hat{f}^2$ of Byzantine clients in this subset.

---

[2] Allouah et al. (2024) uses $b$ to denote the true number of Byzantine clients and $\hat{b}$ to denote the estimation of $b$. We replace them with $f$ and $\hat{f}$ respectively.

However, their convergence requires overestimation of the fraction of Byzantine clients, i.e., $\frac{\hat{f}}{\hat{n}} > \frac{f}{n}$, while the cases of underestimation and exact estimation are not studied. In addition, they obtain the convergence gap $\kappa G^2$, while the optimal lower and upper bound on $\kappa$ is not studied. Yang et al. (2025) studies distributed learning with only $H = 1$ local gradient update and no Byzantine clients ($f = 0$), a small special case of our setting. They found that the performance degrades with $\hat{f}$, which fits our results of non-underestimation in this special case.

## 5 EXPERIMENTS

To demonstrate the aforementioned trade-off in estimating the number of Byzantine clients in federated learning, we apply the FedRo algorithm to a classification task on the CIFAR-10 dataset (Krizhevsky, 2009), using the cross-entropy loss function as the objective function and ResNet-20 (He et al., 2016) as the classifier model.

**Data Assignment:** CIFAR-10 consists of 10 classes and 5k training images per class. We equally divide the 50k samples into $n = 16$ clients. Among the 3125 samples in each client $k$, suppose a fraction $p_{k,c}$ belongs to the $c$-th class. Randomly set $[p_{k,1}, \ldots, p_{k,10}]$ from the 10-dimensional symmetric Dirichlet distribution $\mathrm{Dir}_{10}(\alpha)$. A smaller $\alpha$ corresponds to greater heterogeneity among the clients.

**Models and Training Schemes:** The batch normalization (BN) layers in ResNet-20 are replaced with group normalization (GN) layers, since BN performs poorly with heterogeneous data across clients (Wu & He, 2018). We implement the FedRo algorithm with an $\hat{f}$-CWTM aggregator on a grid of $\hat{f} \in \{0, 1, \ldots, 7\}$, $f \in \{0, 4\}$ and $\alpha \in \{0.1, 1, 10\}$. For $\alpha = 0.1$, we run FedRo for $T = 800$ total communication rounds with $H = 49$ local SGD steps per round. For $\alpha = 1$ and $\alpha = 10$, we run FedRo for $T = 400$ total communication rounds with $H = 98$ local SGD steps per round. Each SGD step uses batchsize 64, weight decay $5 \times 10^{-4}$ and an step-wise diminishing stepsize (see Eq. (46) in Appendix F). Each Byzantine client uploads a vector from normal distribution $\mathcal{N}(0, 5)$. More details on data division, model architectures, and training schemes are provided in Appendix F.

Table 3: Top-1 Accuracies of FedRo Algorithm on CIFAR-10 Data. A smaller $\alpha$ corresponds to higher heterogeneity $G^2$. When $\hat{f} > f$, the accuracy drops compared with the corresponding accuracy under $\hat{f} = f$ (bolded) are marked in the parentheses.

| $\hat{f}$ | $f = 0$ | | | $f = 4$ | | |
|---|---|---|---|---|---|---|
| | $\alpha = 0.1$ | $\alpha = 1.0$ | $\alpha = 10.0$ | $\alpha = 0.1$ | $\alpha = 1.0$ | $\alpha = 10.0$ |
| 0 | **65.61** | **78.76** | **80.24** | 10.01 | 10.11 | 10.01 |
| 1 | 62.28(-3.33) | 77.31(-1.45) | 79.45(-0.79) | 12.85 | 10.04 | 10.00 |
| 2 | 53.01(-12.60) | 77.13(-1.63) | 79.63(-0.61) | 9.17 | 10.00 | 9.85 |
| 3 | 51.87(-13.74) | 76.76(-2.00) | 78.28(-1.96) | 11.46 | 23.98 | 25.70 |
| 4 | 48.47(-17.14) | 76.80(-1.96) | 77.93(-2.31) | **54.58** | **74.19** | **75.80** |
| 5 | 47.25(-18.36) | 76.93(-1.83) | 77.67(-2.57) | 49.83(-4.75) | 73.53(-0.66) | 75.77(-0.03) |
| 6 | 45.27(-20.34) | 75.32(-3.44) | 77.91(-2.33) | 47.65(-6.93) | 72.62(-1.57) | 74.35(-1.45) |
| 7 | 43.38(-22.23) | 75.01(-3.75) | 77.87(-2.37) | 43.69(-10.89) | 72.45(-1.74) | 74.02(-1.78) |

**Main Results:** Table 3 reports the top-1 accuracies for each candidate $(\hat{f}, f, \alpha)$. When $\hat{f} < f = 4$ (underestimation), the accuracies are extremely low, which fits the divergence result in Theorem 3. When $\hat{f} \geq f$ (non-underestimation) in both no-Byzantine ($f = 0$) and Byzantine ($f = 4$) settings, the accuracy decreases in general as $\hat{f}$ increases from $f$. Moreover, with larger $\alpha$ (i.e. lower heterogeneity $G^2$), such an accuracy decrease with larger $\hat{f}$ slows down. These results fit Theorems 4 and 5 which indicate that as $\hat{f}$ increases from $f$, the tight convergence lower bound $\frac{\hat{f}G^2}{n-f-\hat{f}}$ increases at a rate proportional to the heterogeneity $G^2$.

## 6 CONCLUSION

To our knowledge, this is the first work that systematically investigates the theoretical effect of the estimated number of Byzantine clients on both aggregation error and federated learning performance. Both theoretical and empirical results demonstrate that while an aggregator with a larger robustness degree can tolerate more Byzantine clients, the performance can deteriorate when there are actually fewer or even no Byzantine clients.

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

# Appendix

## Table of Contents

## A  PROOF OF THEOREM 1

The commonly used coordinate-wise trimmed mean (CWTM) aggregator $TM_{\hat{f}/n}$ defined by Eq. (5) has been proved to be an $(\hat{f}, \hat{\kappa})$-robust aggregator (see Proposition 2 of Allouah et al. (2023)) with $\hat{\kappa} = \frac{6\hat{f}}{n-2\hat{f}}\left(1 + \frac{\hat{f}}{n-2\hat{f}}\right)$. Hence, it remains to prove that $TM_{\hat{f}/n}$ is not $(f, \kappa)$-robust for any $\hat{f} < f < \frac{n}{2}$ and $\kappa > 0$.

Select the scalars $\{x_k\}_{k=1}^n \subset \mathbb{R}$ as follows.

$$x_1 = x_2 = \cdots = x_{n-f} = 0, x_{n-f+1} = x_{n-f+2} = \cdots = x_n = 1. \tag{16}$$

For $S = [n - f] = \{1, 2, \ldots, n - f\}$ with $|S| = n - f$, we have

$$\overline{x}_S = \frac{1}{|S|} \sum_{k \in S} x_k = 0, \quad \frac{1}{|S|} \sum_{k \in S} (x_k - \overline{x}_S)^2 = 0.$$

Suppose $TM_{\hat{f}/n}$ is $(f, \kappa)$-robust for some $\kappa > 0$, which implies that

$$\left| TM_{\hat{f}/n}(x_1, \ldots, x_n) - \overline{x}_S \right| \le \kappa' \cdot \frac{1}{|S|} \sum_{k \in S} (x_k - \overline{x}_S)^2 = 0, \tag{17}$$

so $TM_{\hat{f}/n}(x_1, \ldots, x_n) = \overline{x}_S = 0$. This contradicts with the definition (5) of $TM_{\hat{f}/n}$ which along with the scalars (16) implies that

$$TM_{\hat{f}/n}(x_1, \ldots, x_n) = \frac{1}{n - 2\hat{f}} \sum_{k=\hat{f}+1}^{n-\hat{f}} x_k = \frac{f - \hat{f}}{n - 2\hat{f}} > 0. \tag{18}$$

Therefore, $TM_{\hat{f}/n}$ is not $(f, \kappa)$-robust for any $\hat{f} < f < \frac{n}{2}$ and $\kappa > 0$.

## B  PROOF OF THEOREM 2

### B.1  PROOF OF ITEM 1

The conclusion is obvious when $f = \hat{f}$, so we consider the case where $0 < f < \hat{f}$.

For any set $S \subset [n]$ with size $|S| = n - f$, $S$ contains $q = \frac{(n-f)!}{(n-\hat{f})!(\hat{f}-f)!}$ subsets $S_1, \ldots, S_q \subset S$ with the same size $|S_1| = \ldots = |S_q| = n - \hat{f}$. Then for any $x_1, \ldots, x_n \in \mathbb{R}^d$ and $E \subset [n]$, denote $\overline{x}_E = \frac{1}{|E|} \sum_{k \in E} x_k$. Then, we prove that $\mathcal{A}$ is $(f, \hat{\kappa})$-robust as follows.

$$\| \mathcal{A}(x_1, \ldots, x_n) - \overline{x}_S \|^2$$

$$= \left\| \mathcal{A}(x_1, \ldots, x_n) - \frac{1}{q} \sum_{i=1}^{q} \overline{x}_{S_i} \right\|^2$$

$$\overset{(a)}{\leq} \frac{1}{q} \sum_{i=1}^{q} \| \mathcal{A}(x_1, \ldots, x_n) - \overline{x}_{S_i} \|^2$$

$$\overset{(b)}{\leq} \frac{1}{q} \sum_{i=1}^{q} \frac{\hat{\kappa}}{n - \hat{f}} \sum_{k \in S_i} \| x_k - \overline{x}_{S_i} \|^2$$

$$= \frac{\hat{\kappa}}{q(n - \hat{f})} \sum_{i=1}^{q} \sum_{k \in S_i} \| (x_k - \overline{x}_S) - (\overline{x}_{S_i} - \overline{x}_S) \|^2$$

$$= \frac{\hat{\kappa}}{q(n - \hat{f})} \sum_{i=1}^{q} \sum_{k \in S_i} \left[ \| x_k - \overline{x}_S \|^2 + \| \overline{x}_{S_i} - \overline{x}_S \|^2 - 2 \langle x_k - \overline{x}_S, \overline{x}_{S_i} - \overline{x}_S \rangle \right]$$

$$= \frac{\hat{\kappa}}{q(n - \hat{f})} \sum_{i=1}^{q} \sum_{k \in S_i} \left[ \| x_k - \overline{x}_S \|^2 \right] + \frac{\hat{\kappa}}{q} \sum_{i=1}^{q} \left[ \| \overline{x}_{S_i} - \overline{x}_S \|^2 \right]$$

$$- \frac{2\hat{\kappa}}{q(n - \hat{f})} \sum_{i=1}^{q} \langle |S_i| \overline{x}_{S_i} - |S_i| \overline{x}_S, \overline{x}_{S_i} - \overline{x}_S \rangle$$

$$\overset{(c)}{=} \frac{\hat{\kappa}(n - \hat{f})!(\hat{f} - f)!}{(n - \hat{f})(n - f)!} \frac{(n - f - 1)!}{(n - \hat{f} - 1)!(\hat{f} - f)!} \sum_{k \in S} \left[ \| x_k - \overline{x}_S \|^2 \right] + \frac{\hat{\kappa}}{q} \sum_{i=1}^{q} \left[ \| \overline{x}_{S_i} - \overline{x}_S \|^2 \right]$$

$$- \frac{2\hat{\kappa}}{q} \sum_{i=1}^{q} \left[ \| \overline{x}_{S_i} - \overline{x}_S \|^2 \right]$$

$$\leq \frac{\hat{\kappa}}{n - f} \sum_{k \in S} \left[ \| x_k - \overline{x}_S \|^2 \right].$$

where (a) applies Jensen's inequality to the convex function $\| \cdot \|^2$, (b) applies the $(\hat{f}, \hat{\kappa})$-robust aggregator $\mathcal{A}$ to the set $S_i$ of size $|S_i| = n - \hat{f}$, (c) uses $q = \frac{(n-f)!}{(n-\hat{f})!(\hat{f}-f)!}$ and the fact that each $k \in S$ is contained by $\frac{(n-f-1)!}{(n-\hat{f}-1)!(\hat{f}-f)!}$ sets of $\{S_i\}_{i=1}^{q}$ (since these sets for a certain $k \in S$ can be obtained by removing $(\hat{f} - f)$ elements from $S \backslash \{k\}$ with size $|S \backslash \{k\}| = n - f - 1$).

## B.2 PROOF OF ITEM 2

Suppose an aggregator $\mathcal{A} : (\mathbb{R}^d)^n \to \mathbb{R}^d$ is $(\hat{f}, \hat{\kappa})$-robust and $(f, \kappa)$-robust. Select the scalars $\{x_k\}_{k=1}^n \subset \mathbb{R}$ as follows.

$$x_1 = x_2 = \ldots = x_{n-\hat{f}} = 0, x_{n-\hat{f}+1} = \ldots = x_n = 1. \tag{19}$$

Since $\mathcal{A}$ is $(\hat{f}, \hat{\kappa})$-robust, for $S' = [n - \hat{f}] = \{1, 2, \ldots, n - \hat{f}\}$ we have

$$\overline{x}_{S'} = \frac{1}{|S'|} \sum_{k \in S'} x_k = 0, \quad |\mathcal{A}(x_1, \ldots, x_n) - \overline{x}_{S'}|^2 \leq \frac{\kappa}{|S'|} \sum_{k \in S'} |x_k - \overline{x}_{S'}|^2 = 0,$$

so $\mathcal{A}(x_1, \ldots, x_n) = \overline{x}_{S'} = 0$.

Denote the set $S = \{f + 1, f + 2, \ldots, n\}$ which contains $n - \hat{f} + 1, \ldots, n$ as $0 \leq f \leq \hat{f} < \frac{n}{2}$, so $\overline{x}_S = \frac{1}{|S|} \sum_{k \in S} x_k = \frac{\hat{f}}{n-f}$. Then since $|S| = n - f$ and $\mathcal{A}$ is $(f, \kappa)$-robust, we have

$$|\mathcal{A}(x_1, \ldots, x_n) - \overline{x}_S|^2 \leq \frac{\kappa}{|S|} \sum_{k \in S} |x_k - \overline{x}_S|^2.$$

Substituting $\mathcal{A}(x_1, \ldots, x_n) = 0, \overline{x}_S = \frac{\hat{f}}{n-f}, |S| = n - f$ and Eq. (19) into the inequality above, we have

$$\begin{aligned}
\frac{\hat{f}^2}{(n-f)^2} &\leq \frac{\kappa}{n-f} \left[ \hat{f} \left( 1 - \frac{\hat{f}}{n-f} \right)^2 + (n - f - \hat{f}) \left( \frac{\hat{f}}{n-f} \right)^2 \right] \\
&= \frac{\kappa}{(n-f)^3} \left[ \hat{f}(n - f - \hat{f})^2 + (n - f - \hat{f})\hat{f}^2 \right] \\
&= \frac{\kappa \hat{f}(n - f - \hat{f})}{(n-f)^3} [(n - f - \hat{f}) + \hat{f}] \\
&= \frac{\kappa \hat{f}(n - f - \hat{f})}{(n-f)^2},
\end{aligned}$$

which implies $\kappa \geq \frac{\hat{f}}{n-f-\hat{f}}$.

## B.3 PROOF OF ITEM 3

We use the composite aggregator $\hat{f}$-Krum$\circ \hat{f}$-NNM is defined by Definition 3 with $\mathcal{A} = \hat{f}$-Krum. Based on Lemmas 1 and 2, for any $0 \leq f \leq \hat{f} < \frac{n}{2}$, $(\hat{f}$-Krum$) \circ (\hat{f}$-NNM$)$ is an $(f, \kappa)$-robust aggregator with

$$\kappa = \frac{12\hat{f}}{n-f} \left( \frac{6(n-f)}{n-f-\hat{f}} + 1 \right) \leq \frac{12\hat{f}}{n-f} \cdot \left( \frac{6(n-f)}{n-f-\hat{f}} + \frac{n-f}{n-f-\hat{f}} \right) = \frac{84\hat{f}}{n-f-\hat{f}}.$$

This concludes the proof of Item 3. It remains to prove Lemmas 1 and 2 in the next two subsections.

## B.4 PROOF OF LEMMA 1 FOR NNM

For any $n \geq 1$, $\hat{f} \in \left[ 0, \frac{n}{2} \right)$, $k \in [n]$ and $x_1, \ldots, x_n \in \mathbb{R}^d$, denote $\mathcal{N}_k \subset [n]$ as the set of $(n - \hat{f})$ indexes from $[n]$ such that $\{x_i\}_{i \in \mathcal{N}_k}$ are the $(n - \hat{f})$ nearest neighbors of $x_k$. In other words, $\max_{i \in \mathcal{N}_k} \|x_i - x_k\| \leq \min_{j \in [n] \setminus \mathcal{N}_k} \|x_j - x_k\|$. Then, for any set $S \subset [n]$ with size $|S| = n - f$, we have

$$\frac{1}{|\mathcal{N}_k|} \sum_{i \in \mathcal{N}_k} \|x_i - x_k\|^2 \leq \frac{1}{|S|} \sum_{i \in S} \|x_i - x_k\|^2, \tag{20}$$

since $\frac{1}{|\mathcal{N}_k|} \sum_{i \in \mathcal{N}_k} \|x_k - x_i\|^2$ is the average of the $|\mathcal{N}_k| = n - \hat{f}$ smallest numbers in $\{\|x_k - x_i\|^2\}_{i=1}^n$, while $\frac{1}{|S|} \sum_{i \in S} \|x_k - x_i\|^2$ is the average of $|S| = n - f$ ($|S| \geq |\mathcal{N}_k|$) numbers in $\{\|x_k - x_i\|^2\}_{i=1}^n$. Then we have

$$\|y_k - \overline{x}_S\|^2$$

$$= \left\| \frac{1}{n-\hat{f}} \sum_{i \in \mathcal{N}_k} x_i - \frac{1}{n-f} \sum_{i \in S} x_i \right\|^2$$

$$= \left\| \left( \frac{1}{n-\hat{f}} - \frac{1}{n-f} \right) \sum_{i \in S \cap \mathcal{N}_k} (x_i - x_k) + \frac{1}{n-\hat{f}} \sum_{i \in \mathcal{N}_k \backslash S} (x_i - x_k) - \frac{1}{n-f} \sum_{i \in S \backslash \mathcal{N}_k} (x_i - x_k) \right\|^2$$

$$\leq 3 \left\| \frac{\hat{f}-f}{(n-\hat{f})(n-f)} \sum_{i \in S \cap \mathcal{N}_k} (x_i - x_k) \right\|^2 + 3 \left\| \frac{1}{n-\hat{f}} \sum_{i \in \mathcal{N}_k \backslash S} (x_i - x_k) \right\|^2$$

$$+ 3 \left\| -\frac{1}{n-f} \sum_{i \in S \backslash \mathcal{N}_k} (x_i - x_k) \right\|^2$$

$$\leq \frac{3|S \cap \mathcal{N}_k|(\hat{f}-f)^2}{(n-\hat{f})^2(n-f)^2} \sum_{i \in S \cap \mathcal{N}_k} \|x_i - x_k\|^2 + \frac{3|\mathcal{N}_k \backslash S|}{(n-\hat{f})^2} \sum_{i \in \mathcal{N}_k \backslash S} \|x_i - x_k\|^2$$

$$+ \frac{3|S \backslash \mathcal{N}_k|}{(n-f)^2} \sum_{i \in S \backslash \mathcal{N}_k} \|x_i - x_k\|^2$$

$$\overset{(a)}{\leq} \frac{3(\hat{f}-f)^2}{(n-\hat{f})(n-f)^2} \sum_{i \in S \cap \mathcal{N}_k} \|x_i - x_k\|^2 + \frac{3f}{(n-\hat{f})^2} \sum_{i \in \mathcal{N}_k} \|x_i - x_k\|^2$$

$$+ \frac{3\hat{f}}{(n-f)^2} \sum_{i \in S \backslash \mathcal{N}_k} \|x_i - x_k\|^2$$

$$\overset{(b)}{\leq} \frac{3\hat{f}}{|S|(n-f)} \sum_{i \in S \cap \mathcal{N}_k} \|x_i - x_k\|^2 + \frac{3f|\mathcal{N}_k|}{|S|(n-\hat{f})^2} \sum_{i \in \mathcal{S}} \|x_i - x_k\|^2 + \frac{3\hat{f}}{|S|(n-f)} \sum_{i \in S \backslash \mathcal{N}_k} \|x_i - x_k\|^2$$

$$\overset{(c)}{\leq} \left[ \frac{3\hat{f}}{|S|(n-f)} + \frac{3f}{|S|(n-\hat{f})} \right] \sum_{i \in S} \|x_i - x_k\|^2$$

$$\overset{(d)}{\leq} \frac{6\hat{f}}{|S|(n-f)} \sum_{i \in S} \|x_i - x_k\|^2, \tag{21}$$

where (a) uses $|S \cap \mathcal{N}_k| \leq |\mathcal{N}_k| = n - \hat{f}$, $|\mathcal{N}_k \backslash S| \leq n - |S| = f$, $|S \backslash \mathcal{N}_k| \leq n - |\mathcal{N}_k| = \hat{f}$, (b) uses Eq. (20), $0 \leq f \leq \hat{f} < \frac{n}{2}$ (so $\hat{f} - f \leq \hat{f} \leq n - \hat{f}$) and $|S| = n - f$, (c) uses $|\mathcal{N}_k| = n - \hat{f}$, and (d) uses the following inequality.

$$\frac{3\hat{f}}{|S|(n-f)} - \frac{3f}{|S|(n-\hat{f})} = \frac{3\hat{f}(n-\hat{f}) - 3f(n-f)}{|S|(n-f)(n-\hat{f})} = \frac{3(\hat{f}-f)(n-\hat{f}-f)}{|S|(n-f)(n-\hat{f})} \geq 0.$$

Denote $(y_1, \ldots, y_n) = \hat{f}\text{-NNM}(x_1, \ldots, x_n)$ where $y_k = \frac{1}{n-\hat{f}} \sum_{i \in \mathcal{N}(x_k)} x_i$ as defined in Eq. (6). Then denote $\overline{x}_S = \frac{1}{S} \sum_{k \in S} x_k$ and $\overline{y}_S = \frac{1}{S} \sum_{k \in S} y_k$. We obtain that

$$\|\overline{y}_S - \overline{x}_S\|^2 + \frac{1}{|S|} \sum_{k \in S} \|y_k - \overline{y}_S\|^2$$

$$= \frac{1}{|S|} \sum_{k \in S} \left( \|y_k - \overline{y}_S\|^2 + \|\overline{y}_S - \overline{x}_S\|^2 \right)$$

$$= \frac{1}{|S|} \sum_{k \in S} \left[ \|(y_k - \overline{y}_S) + (\overline{y}_S - \overline{x}_S)\|^2 - 2\langle y_k - \overline{y}_S, \overline{y}_S - \overline{x}_S \rangle \right]$$

$$= \frac{1}{|S|} \sum_{k \in S} \|y_k - \overline{x}_S\|^2 - 2 \left\langle \overline{y}_S - \overline{x}_S, \frac{1}{|S|} \sum_{k \in S} (y_k - \overline{y}_S) \right\rangle$$

$$\overset{(a)}{\leq} \frac{1}{|S|} \sum_{k \in S} \frac{6\hat{f}}{|S|(n-f)} \sum_{i \in S} \|x_i - x_k\|^2 - 2\langle \overline{y}_S - \overline{x}_S, \overline{y}_S - \overline{y}_S \rangle$$

$$=\frac{6\hat{f}}{|S|^2(n-f)}\sum_{i,k\in S}\left(\|x_i-\overline{x}_S\|^2+\|x_k-\overline{x}_S\|^2-2\langle x_i-\overline{x}_S,x_k-\overline{x}_S\rangle\right)$$

$$=\frac{6\hat{f}}{|S|^2(n-f)}\left[|S|\sum_{i\in S}(\|x_i-\overline{x}_S\|^2)+|S|\sum_{k\in S}(\|x_k-\overline{x}_S\|^2)-2\langle|S|\overline{x}_S-|S|\overline{x}_S,|S|\overline{x}_S-|S|\overline{x}_S\rangle\right]$$

$$=\frac{12\hat{f}}{n-f}\cdot\frac{1}{|S|}\sum_{i\in S}\|x_i-\overline{x}_S\|^2, \tag{22}$$

where (a) uses Eq. (21).

For any $S\subset[n]$ with size $|S|=n-f$, we prove below that the composite aggregator $\mathcal{A}\circ\hat{f}\text{-NNM}(x_1,\ldots,x_n)=\mathcal{A}(y_1,\ldots,y_n)$ is $(f,\kappa)$-robust with $\kappa=\frac{12\hat{f}(1+\kappa)}{n-f}$.

$$\|\mathcal{A}\circ\hat{f}\text{-NNM}(x_1,\ldots,x_n)-\overline{x}_S\|^2$$

$$\leq(1+\kappa^{-1})\|\mathcal{A}(y_1,\ldots,y_n)-\overline{y}_S\|^2+(1+\kappa)\|\overline{y}_S-\overline{x}_S\|^2$$

$$\overset{(a)}{\leq}\frac{1+\kappa}{|S|}\sum_{k=1}^{n}\|y_k-\overline{y}_S\|^2+(1+\kappa)\|\overline{y}_S-\overline{x}_S\|^2$$

$$\overset{(b)}{\leq}\frac{12\hat{f}(1+\kappa)}{n-f}\cdot\frac{1}{|S|}\sum_{i\in S}\|x_i-\overline{x}_S\|^2 \tag{23}$$

where (a) uses the fact that $\mathcal{A}$ is an $(f,\kappa)$-robust aggregator, (b) uses Eq. (22).

### B.5 PROOF OF LEMMA 2 FOR KRUM AGGREGATOR

For any $n\geq 1$, $\hat{f}\in\left[0,\frac{n}{2}\right)$, $k\in[n]$ and $x_1,\ldots,x_n\in\mathbb{R}^d$, denote $\mathcal{N}_k\subset[n]$ as the set of $(n-\hat{f})$ indexes from $[n]$ such that $\{x_i\}_{i\in\mathcal{N}_k}$ are the $(n-\hat{f})$ nearest neighbors of $x_k$. In other words, $\max_{i\in\mathcal{N}_k}\|x_i-x_k\|\leq\min_{j\in[n]\setminus\mathcal{N}_k}\|x_j-x_k\|$. Then, for any set $S\subset[n]$ with size $|S|=n-f$, Eq. (20) has been proved as repeated below.

$$\frac{1}{|\mathcal{N}_k|}\sum_{i\in\mathcal{N}_k}\|x_k-x_i\|^2\leq\frac{1}{|S|}\sum_{i\in\mathcal{S}}\|x_k-x_i\|^2. \tag{24}$$

Then for $k^*\in[n]$ defined in the $\hat{f}$-Krum (7), we have

$$\sum_{i\in\mathcal{N}_{k^*}}\|x_{k^*}-x_i\|^2$$

$$=\min_{k\in[n]}\sum_{i\in\mathcal{N}_k}\|x_k-x_i\|^2$$

$$\leq\frac{1}{|S|}\sum_{k\in S}\sum_{i\in\mathcal{N}_k}\|x_k-x_i\|^2$$

$$\overset{(a)}{\leq}\frac{n-\hat{f}}{|S|^2}\sum_{i,k\in S}\|x_k-\overline{x}_S-(x_i-\overline{x}_S)\|^2$$

$$=\frac{n-\hat{f}}{|S|^2}\sum_{i,k\in S}\left(\|x_k-\overline{x}_S\|^2+\|x_i-\overline{x}_S\|^2-2\langle x_k-\overline{x}_S,x_i-\overline{x}_S\rangle\right)$$

$$=\frac{n-\hat{f}}{|S|^2}\left[\sum_{i,k\in S}\|x_k-\overline{x}_S\|^2+\sum_{i,k\in S}\|x_i-\overline{x}_S\|^2-2\sum_{i,k\in S}\langle x_k-\overline{x}_S,x_i-\overline{x}_S\rangle\right]$$

$$=\frac{n-\hat{f}}{|S|^2}\left[2|S|\sum_{i\in S}\|x_i-\overline{x}_S\|^2-2\sum_{i\in S}\Big\langle\underbrace{\sum_{k\in S}(x_k-\overline{x}_S)}_{=0},x_i-\overline{x}_S\Big\rangle\right]$$

$$\overset{(b)}{\leq} 2 \sum_{i \in S} \|x_i - \overline{x}_S\|^2. \tag{25}$$

where (a) uses Eq. (24) and $|\mathcal{N}_k| = n - \hat{f}$, and (b) uses $|S| = n - f \geq n - \hat{f}$. Note that

$$\|x_{k^*} - \overline{x}_S\|^2 \leq 2\|x_{k^*} - x_i\|^2 + 2\|x_i - \overline{x}_S\|^2, \forall i \in S \tag{26}$$

which can be rearranged into

$$\|x_{k^*} - x_i\|^2 \geq \frac{1}{2}\|x_{k^*} - \overline{x}_S\|^2 - \|x_i - \overline{x}_S\|^2, \forall i \in S. \tag{27}$$

Then, we have

$$\sum_{i \in \mathcal{N}_{k^*}} \|x_{k^*} - x_i\|^2 \geq \sum_{i \in S \cap \mathcal{N}_{k^*}} \|x_{k^*} - x_i\|^2$$

$$\overset{(a)}{\geq} \frac{|S \cap \mathcal{N}_{k^*}|}{2}\|x_{k^*} - \bar{x}_S\|^2 - \sum_{i \in S \cap \mathcal{N}_{k^*}} \|x_i - \bar{x}_S\|^2$$

$$\overset{(b)}{\geq} \frac{n - f - \hat{f}}{2}\|x_{k^*} - \bar{x}_S\|^2 - \sum_{i \in S \cap \mathcal{N}_{k^*}} \|x_i - \bar{x}_S\|^2,$$

where (a) uses Eq. (27) and (b) uses $|S \cap \mathcal{N}_{k^*}| = |S| + |\mathcal{N}_{k^*}| - |S \cup \mathcal{N}_{k^*}| \geq (n - f) + (n - \hat{f}) - n = n - f - \hat{f}$. By rearranging the inequality above, we have

$$\|x_{k^*} - \bar{x}_S\|^2 \leq \frac{2}{n - f - \hat{f}}\Big[\sum_{i \in \mathcal{N}_{k^*}} \|x_{k^*} - x_i\|^2 + \sum_{i \in S \cap \mathcal{N}_{k^*}} \|x_i - \bar{x}_S\|^2\Big]$$

$$\overset{(a)}{\leq} \frac{2}{n - f - \hat{f}}\Big[2\sum_{i \in S} \|x_i - \overline{x}_S\|^2 + \sum_{i \in S} \|x_i - \overline{x}_S\|^2\Big]$$

$$\leq \frac{6}{n - f - \hat{f}}\sum_{i \in S} \|x_i - \overline{x}_S\|^2 = \frac{6(n - f)}{n - f - \hat{f}} \cdot \frac{1}{|S|}\sum_{i \in S} \|x_i - \overline{x}_S\|^2,$$

where (a) uses Eq. (25). Therefore, $(\hat{f}\text{-Krum})(x_1, \ldots, x_n) = x_{k^*}$ defined by Eq. (7) is $(f, \kappa)$-robust with $\kappa \leq \frac{6(n-f)}{n-f-\hat{f}}$

## C  PROOF OF THEOREM 3

Without loss of generality, use $\mathcal{H} = [n - f] = \{1, 2, \ldots, n - f\}$ as the set of honest clients. Select the loss function $\ell_k(w) = \frac{w^2}{2}(w \in \mathbb{R})$ for every client $k \in [n]$, which satisfies Assumptions 1-4.

In Algorithm 1, use the coordinate-wise trimmed mean (CWTM) aggregator defined by Eq. (5) which has been proved to be an $(\hat{f}, \hat{\kappa})$-robust aggregator (see Proposition 2 of Allouah et al. (2023)) with $\hat{\kappa} = \frac{6\hat{f}}{n - 2\hat{f}}\big(1 + \frac{\hat{f}}{n - 2\hat{f}}\big)$. All the honest clients $k \in \mathcal{H} = [n - f]$ perform the local gradient descent updates (2) as follows.

$$w_{t,h+1}^{(k)} = w_{t,h}^{(k)} - \gamma \ell_k'(w_{t,h}^{(k)}) = (1 - \gamma)w_{t,h}^{(k)}, k \in \mathcal{H} = [n - f]. \tag{28}$$

Iterating the update rule above yields that

$$w_t^{(k)} = w_{t,H}^{(k)} = (1 - \gamma)^H w_{t,0}^{(k)} = (1 - \gamma)^H w_t, k \in \mathcal{H} = [n - f].$$

In contrast, we let the Byzantine clients $k \in [n]\backslash\mathcal{H} = \{n - f + 1, n - f + 2, \ldots, n\}$ upload $w_t^{(k)} = n|(1 - \gamma)^H w_t| + t$. Therefore, the global parameter update rule (3) becomes

$$w_{t+1} = w_t + \mathcal{A}(\{w_t^{(k)} - w_t\}_{k=1}^n)$$

$$= w_t + \frac{(n - f - \hat{f})[(1 - \gamma)^H w_t - w_t] + (f - \hat{f})[n|(1 - \gamma)^H w_t| - w_t + t]}{n - 2\hat{f}}$$

$$= \frac{n(f - \hat{f})|(1 - \gamma)^H w_t| - (n - f - \hat{f})|(1 - \gamma)^H w_t| + t(f - \hat{f})}{n - 2\hat{f}}$$

$$\overset{(a)}{\geq} \frac{t(f - \hat{f})}{n - 2\hat{f}},$$

where (a) uses the CWTM aggregator $\mathcal{A}$ defined by Eq. (5) as well as the fact that $\{w_t^{(k)} - w_t\}_{k=1}^n$ contains $n - f$ scalars $(1 - \gamma)^H w_t - w_t$ and $f$ larger scalars $n|(1 - \gamma)^H w_t| - w_t + t$, and (b) uses $n(f - \hat{f}) - (n - f - \hat{f}) > f + \hat{f} > 0$ and $n - 2\hat{f} > 0$ since the integers $\hat{f}$, $f$, $n$ satisfy $0 \leq \hat{f} < f < \frac{n}{2}$. Since $\frac{(f - \hat{f})}{n - 2\hat{f}} > 0$, the inequality above implies that $|w_t| \to +\infty$ as $t \to +\infty$. Note that the objective function (1) in this example is $\ell_{\mathcal{H}}(w) = \frac{w^2}{2}$. Hence, Eqs. (9) and (10) can be proved as follows as $T \to +\infty$.

$$\frac{1}{T} \sum_{t=0}^{T-1} |\ell'_{\mathcal{H}}(w_t)|^2 = \frac{1}{T} \sum_{t=0}^{T-1} |w_t|^2 \to +\infty.$$

$$\ell_{\mathcal{H}}(w_T) - \ell^* = \frac{w_T^2}{2} - 0 \to +\infty.$$

## D    PROOF OF THEOREM 4

Without loss of generality, use $\mathcal{H} = [n - f] = \{1, 2, \ldots, n - f\}$. Let the Byzantine clients adopt the same honest behavior as the honest clients, i.e., upload the result after $H$ local gradient descent updates. This is valid since Byzantine clients can upload any vectors.

Select the following loss functions.

$$\ell_k(w) = \begin{cases} cG(w + 1)^2, k = 1, 2, \ldots, \hat{f} \\ cGw^2, k = \hat{f} + 1, \hat{f} + 2, \ldots, n \end{cases}, \tag{29}$$

where $w \in \mathbb{R}$ and the constant $c > 0$ is to be selected later. The derivatives of these loss functions are shown below.

$$\ell'_k(w) = \begin{cases} 2cG(w + 1), k = 1, 2, \ldots, \hat{f} \\ 2cGw, k = \hat{f} + 1, \hat{f} + 2, \ldots, n \end{cases}. \tag{30}$$

It is straightforward to check that these loss functions (29) satisfy Assumption 2, i.e., $L$-smoothness with Lipschitz constant $L = 2cG$. Then the objective function (1) is

$$\ell_{\mathcal{H}}(w) = \frac{1}{|\mathcal{H}|} \sum_{k \in \mathcal{H}} \ell_k(w)$$

$$= \frac{1}{n - f} [\hat{f} \cdot cG(w + 1)^2 + (n - f - \hat{f}) \cdot cGw^2]$$

$$= \frac{cG}{n - f} \left[ (n - f)w^2 + 2\hat{f}w + \hat{f} \right]$$

$$= cG \left[ \left( w + \frac{\hat{f}}{n - f} \right)^2 + \frac{\hat{f}(n - f - \hat{f})}{(n - f)^2} \right], \tag{31}$$

which has the following derivative.

$$\ell'_{\mathcal{H}}(w) = 2cG \left( w + \frac{\hat{f}}{n - f} \right), \tag{32}$$

The objective function (31) satisfies Assumption 1 with $\ell^* = \inf_{w \in \mathbb{R}^d} \ell_{\mathcal{H}}(w) = \frac{cG\hat{f}(n - f - \hat{f})}{(n - f)^2}$, and also satisfies Assumption 4 with $\mu = 2cG$ since

$$\ell_{\mathcal{H}}(w) - \ell^* - \frac{1}{2\mu} \|\nabla \ell_{\mathcal{H}}(w)\|^2 = cG \left( w + \frac{\hat{f}}{n - f} \right)^2 - \frac{1}{4cG} \cdot 4c^2 G^2 \left( w + \frac{\hat{f}}{n - f} \right)^2 = 0.$$

Next, we will check Assumption 3 as follows.

$$\frac{1}{|\mathcal{H}|} \sum_{k \in \mathcal{H}} |\ell'_k(w) - \ell'_{\mathcal{H}}(w)|^2$$

$$= \frac{1}{n-f} \Big[ \hat{f} \cdot (2cG)^2 \Big( \frac{n-f-\hat{f}}{n-f} \Big)^2 + (n-f-\hat{f}) \cdot (2cG)^2 \Big( \frac{\hat{f}}{n-f} \Big)^2 \Big]$$

$$= \frac{(2cG)^2}{(n-f)^3} \big[ \hat{f}(n-f-\hat{f})^2 + (n-f-\hat{f})\hat{f}^2 \big]$$

$$= \frac{\hat{f}(n-f-\hat{f})(2cG)^2}{(n-f)^2} \tag{33}$$

By letting the equation above equal to $G^2$ to ensure Assumption 3, we can select

$$c = \frac{n-f}{2\sqrt{\hat{f}(n-f-\hat{f})}}. \tag{34}$$

Suppose all the Byzantine clients behave in the same way as the honest clients. Then every client $k \in S := \{\hat{f}+1, \ldots, n\}$ adopts the following local gradient descent update.

$$w_{t,h+1}^{(k)} = w_{t,h}^{(k)} - \gamma \ell'_k(w_{t,h}^{(k)}) = (1 - 2cG\gamma) w_{t,h}^{(k)}, \forall k \in S.$$

Iterating the update rule above yields that

$$w_t^{(k)} = w_{t,H}^{(k)} = (1 - 2cG\gamma)^H w_{t,0}^{(k)} = (1 - 2cG\gamma)^H w_t, \forall k \in S.$$

Since $|S| = n - \hat{f}$, $\frac{1}{|S|} \sum_{k \in S} (w_t^{(k)} - w_t) = [(1 - 2cG\gamma)^H - 1] w_t$ and $\mathcal{A}$ is an $(\hat{f}, \hat{\kappa})$-robust aggregator, we have

$$\big| \mathcal{A}(\{w_t^{(k)} - w_t\}_{k=1}^n) - [(1 - 2cG\gamma)^H - 1] w_t \big|^2$$

$$\leq \hat{\kappa} \cdot \frac{1}{|S|} \sum_{k \in S} \big| w_t^{(k)} - w_t - [(1 - 2cG\gamma)^H - 1] w_t \big|^2 = 0,$$

so the global parameter update rule (3) becomes

$$w_{t+1} = w_t + \mathcal{A}(\{w_t^{(k)} - w_t\}_{k=1}^n) = w_t + [(1 - 2cG\gamma)^H - 1] w_t = (1 - 2cG\gamma)^H w_t. \tag{35}$$

We consider the following cases of hyperparameter choices.

**(Case 1):** When $0 < \gamma < \frac{1}{cG}$, we have $|(1 - 2cG\gamma)^H| < 1$ and thus $w_t \to 0$ by Eq. (35). This implies that as $T \to +\infty$, we have

$$\frac{1}{T} \sum_{t=0}^{T-1} \|\nabla \ell_{\mathcal{H}}(w_t)\|^2 \to |\ell'_{\mathcal{H}}(0)|^2 = \frac{4c^2G^2\hat{f}^2}{(n-f)^2} = \frac{\hat{f}G^2}{n-f-\hat{f}}$$

and

$$\ell_{\mathcal{H}}(w_T) - \ell^* \to \ell_{\mathcal{H}}(0) - \ell^* = cG\Big( \frac{\hat{f}}{n-f} \Big)^2 = \frac{2c^2G^2}{\mu} \Big( \frac{\hat{f}}{n-f} \Big)^2 = \frac{\hat{f}G^2}{2\mu(n-f-\hat{f})}$$

where we use $\mu = 2cG$ and the choice of $c$ in Eq. (34). Hence, Eqs. (11) and (12) hold in this case.

**(Case 2):** When $\gamma = \frac{1}{cG}$ and $H$ is an even number, Eq. (35) implies that $w_t = w_{t-1}$ and thus $w_t \equiv w_0$.

**(Case 3):** When $\gamma = \frac{1}{cG}$ and $H$ is an odd number, Eq. (35) implies that $w_t = -w_{t-1}$. Then for any even number $T$, we have

$$\frac{1}{T} \sum_{t=0}^{T-1} \|\nabla \ell_{\mathcal{H}}(w_t)\|^2 = \frac{1}{2} \big[ |\ell'_{\mathcal{H}}(w_0)|^2 + |\ell'_{\mathcal{H}}(-w_0)|^2 \big]$$

$$=\frac{1}{2}\Big[4c^2G^2\Big(w_0+\frac{\hat{f}}{n-f}\Big)^2+4c^2G^2\Big(-w_0+\frac{\hat{f}}{n-f}\Big)^2\Big]$$

$$=4\cdot\frac{(n-f)^2}{4\hat{f}(n-f-\hat{f})}\cdot G^2\Big(w_0^2+\frac{\hat{f}^2}{(n-f)^2}\Big)\geq\frac{\hat{f}G^2}{n-f-\hat{f}}.$$

and

$$\limsup_{T\to\infty}\ell_H(w_T)-\ell^*=cG\max\Big[\Big(w_0+\frac{\hat{f}}{n-f}\Big)^2,\Big(-w_0+\frac{\hat{f}}{n-f}\Big)^2\Big]$$

$$\geq\frac{2c^2G^2}{\mu}\frac{\hat{f}^2}{(n-f)^2}=\frac{\hat{f}G^2}{2\mu(n-f-\hat{f})},$$

where we use $\mu=2cG$ and the choice of $c$ in Eq. (34). Hence, Eqs. (11) and (12) hold in this case.

**(Case 4):** When $\gamma>\frac{1}{cG}$, we have $|(1-2cG\gamma)^H|>1$ and thus $|w_t|\to+\infty$ by Eq. (35). This implies that as $T\to+\infty$, we have

$$|\ell_{\mathcal{H}}'(w_T)|^2=4c^2G^2\Big(w_T+\frac{\hat{f}}{n-f}\Big)^2\to+\infty\Rightarrow\frac{1}{T}\sum_{t=0}^{T-1}\|\nabla\ell_{\mathcal{H}}(w_t)\|^2\to+\infty$$

and

$$\ell_{\mathcal{H}}(w_T)-\ell^*=cG\Big(w_T+\frac{\hat{f}}{n-f}\Big)^2\to+\infty.$$

Hence, Eqs. (11) and (12) hold in this case.

# E    PROOF OF THEOREM 5

## E.1    SUPPORTING LEMMAS FOR THEOREM 5

**Lemma 3.** *Suppose $L^2\gamma_t^2H(H-1)\leq\frac{1}{2}$ and Assumption 3 holds. Then $V_{t,k}\overset{\text{def}}{=}\sum_{h=0}^{H-1}\|w_{t,h}^{(k)}-w_t\|^2$ obtained from Algorithm 1 has the following upper bounds.*

$$V_{t,k}\leq 2\gamma_t^2H^2(H-1)\|\nabla\ell_k(w_t)\|^2 \tag{36}$$

$$\sum_{k\in\mathcal{H}}V_{t,k}\leq 4\gamma_t^2|\mathcal{H}|H^2(H-1)\big[G^2+\|\nabla\ell_{\mathcal{H}}(w_t)\|^2\big] \tag{37}$$

*Proof.*

$$V_{t,k}\overset{\text{def}}{=}\sum_{h=0}^{H-1}\|w_{t,h}^{(k)}-w_t\|^2$$

$$\overset{(a)}{=}\sum_{h=0}^{H-1}\Big\|\sum_{h'=0}^{h-1}\gamma_t\nabla\ell_k(w_{t,h'}^{(k)})\Big\|^2$$

$$\leq\gamma_t^2\sum_{h=0}^{H-1}\sum_{h'=0}^{h-1}h\|\nabla\ell_k(w_{t,h'}^{(k)})\|^2$$

$$=\gamma_t^2\sum_{h'=0}^{H-2}\sum_{h=h'+1}^{H-1}h\|\nabla\ell_k(w_{t,h'}^{(k)})\|^2$$

$$\overset{(b)}{\leq}\frac{\gamma_t^2H(H-1)}{2}\sum_{h'=0}^{H-1}\|\nabla\ell_k(w_{t,h'}^{(k)})\|^2$$

$$\leq\gamma_t^2H(H-1)\sum_{h=0}^{H-1}\big[\|\nabla\ell_k(w_{t,h}^{(k)})-\nabla\ell_k(w_t)\|^2+\|\nabla\ell_k(w_t)\|^2\big]$$

$$\leq L^2 \gamma_t^2 H(H-1) \sum_{h=0}^{H-1} \left[ \|w_{t,h}^{(k)} - w_t\|^2 \right] + \gamma_t^2 H^2 (H-1) \|\nabla \ell_k(w_t)\|^2$$

$$\overset{(c)}{\leq} \frac{1}{2} V_{t,k} + \gamma_t^2 H^2 (H-1) \|\nabla \ell_k(w_t)\|^2 \tag{38}$$

where (a) uses the local update rule (2), (b) uses $\sum_{h=h'+1}^{H-1} h \leq \sum_{h=1}^{H-1} h = \frac{H(H-1)}{2}$, and (c) uses $L^2 \gamma_t^2 H(H-1) \leq \frac{1}{2}$. Hence, Eq. (36) can be proved by rearranging the inequality above.

Therefore, we can prove Eq. (37) by summing Eq. (36) over $k \in \mathcal{H}$ as follows.

$$\sum_{k \in \mathcal{H}} V_{t,k} \leq 2\gamma_t^2 H^2 (H-1) \sum_{k \in \mathcal{H}} \|\nabla \ell_k(w_t)\|^2$$

$$\leq 4\gamma_t^2 H^2 (H-1) \sum_{k \in \mathcal{H}} \left[ \|\nabla \ell_k(w_t) - \nabla \ell_{\mathcal{H}}(w_t)\|^2 + \|\nabla \ell_{\mathcal{H}}(w_t)\|^2 \right]$$

$$\leq 4\gamma_t^2 |\mathcal{H}| H^2 (H-1) \left[ G^2 + \|\nabla \ell_{\mathcal{H}}(w_t)\|^2 \right],$$

where the final $\leq$ uses Assumption 3. $\qquad\square$

**Lemma 4.** *Define the following quantity obtained from Algorithm 1.*

$$\Delta_t \overset{\text{def}}{=} \frac{1}{|\mathcal{H}|} \sum_{k \in \mathcal{H}} (w_t^{(k)} - w_t) \overset{\text{Eq.(2)}}{=} -\frac{\gamma_t}{|\mathcal{H}|} \sum_{k \in \mathcal{H}} \sum_{h=0}^{H-1} \nabla \ell_k(w_{t,h}^{(k)}). \tag{39}$$

*Suppose $L^2 \gamma_t^2 H(H-1) \leq \frac{1}{2}$, the aggregator used in Algorithm 1 is $(f, \kappa)$-robust and Assumptions 2-3 hold. Then $\Delta_t$ above satisfies the following two bounds.*

$$\|w_{t+1} - w_t - \Delta_t\|^2 \leq 3\kappa H^2 \gamma_t^2 G^2 [1 + 8L^2 H(H-1)\gamma_t^2]$$
$$+ 24\kappa L^2 H^3 (H-1) \gamma_t^4 \|\nabla \ell_{\mathcal{H}}(w_t)\|^2, \tag{40}$$

$$\|\Delta_t + H\gamma_t \nabla \ell_{\mathcal{H}}(w_t)\|^2 \leq 4L^2 \gamma_t^4 H^3 (H-1) \left[ G^2 + \|\nabla \ell_{\mathcal{H}}(w_t)\|^2 \right], \tag{41}$$

*Proof.* Since $\mathcal{A}$ is an $(f, \kappa)$-robust aggregator ($f = n - |\mathcal{H}|$) by Definition 1, we can prove Eq. (40) as follows.

$$\|w_{t+1} - w_t - \Delta_t\|^2$$
$$= \|\mathcal{A}(\{w_t^{(k)} - w_t\}_{k=1}^n) - \Delta_t\|^2$$
$$\leq \frac{\kappa}{|\mathcal{H}|} \sum_{k \in \mathcal{H}} \|w_t^{(k)} - w_t - \Delta_t\|^2$$
$$\overset{(a)}{=} \frac{\kappa}{|\mathcal{H}|} \sum_{k \in \mathcal{H}} \left\| -\gamma_t \sum_{h=0}^{H-1} \nabla \ell_k(w_{t,h}^{(k)}) + \frac{\gamma_t}{|\mathcal{H}|} \sum_{k' \in \mathcal{H}} \sum_{h=0}^{H-1} \nabla \ell_{k'}(w_{t,h}^{(k')}) \right\|^2$$
$$= \frac{\kappa \gamma_t^2}{|\mathcal{H}|} \sum_{k \in \mathcal{H}} \left\| \sum_{h=0}^{H-1} \left[ \nabla \ell_k(w_{t,h}^{(k)}) - \frac{1}{|\mathcal{H}|} \sum_{k' \in \mathcal{H}} \nabla \ell_{k'}(w_{t,h}^{(k')}) \right] \right\|^2$$
$$\leq \frac{\kappa H \gamma_t^2}{|\mathcal{H}|} \sum_{k \in \mathcal{H}} \sum_{h=0}^{H-1} \left\| \nabla \ell_k(w_{t,h}^{(k)}) - \frac{1}{|\mathcal{H}|} \sum_{k' \in \mathcal{H}} \nabla \ell_{k'}(w_{t,h}^{(k')}) \right\|^2$$
$$= \kappa H \gamma_t^2 \sum_{h=0}^{H-1} \frac{1}{|\mathcal{H}|} \sum_{k \in \mathcal{H}} \left\| \nabla \ell_k(w_{t,h}^{(k)}) - \frac{1}{|\mathcal{H}|} \sum_{k' \in \mathcal{H}} \nabla \ell_{k'}(w_{t,h}^{(k')}) \right\|^2$$
$$\leq 3\kappa H \gamma_t^2 \sum_{h=0}^{H-1} \frac{1}{|\mathcal{H}|} \sum_{k \in \mathcal{H}} \left[ \left\| \nabla \ell_k(w_t) - \frac{1}{|\mathcal{H}|} \sum_{k' \in \mathcal{H}} \nabla \ell_{k'}(w_t) \right\|^2 \right.$$
$$+ \left\| \nabla \ell_k(w_{t,h}^{(k)}) - \nabla \ell_k(w_t) \right\|^2 + \left\| \frac{1}{|\mathcal{H}|} \sum_{k' \in \mathcal{H}} [\nabla \ell_{k'}(w_t) - \nabla \ell_{k'}(w_{t,h}^{(k')})] \right\|^2 \right]$$

$$\overset{(b)}{\leq} 3\kappa H^2 \gamma_t^2 G^2 + \frac{3L^2 \kappa H \gamma_t^2}{|\mathcal{H}|} \sum_{h=0}^{H-1} \sum_{k \in \mathcal{H}} \|w_{t,h}^{(k)} - w_t\|^2$$

$$+ 3\kappa H \gamma_t^2 \sum_{h=0}^{H-1} \frac{1}{|\mathcal{H}|} \sum_{k' \in \mathcal{H}} \|\nabla \ell_{k'}(w_t) - \nabla \ell_{k'}(w_{t,h}^{(k')})\|^2$$

$$\overset{(c)}{\leq} 3\kappa H^2 \gamma_t^2 G^2 + \frac{6L^2 \kappa H \gamma_t^2}{|\mathcal{H}|} \sum_{k \in \mathcal{H}} V_{t,k}$$

$$\overset{(d)}{\leq} 3\kappa H^2 \gamma_t^2 G^2 + \frac{6L^2 \kappa H \gamma_t^2}{|\mathcal{H}|} \cdot 4\gamma_t^2 |\mathcal{H}| H^2 (H-1) \big[G^2 + \|\nabla \ell_{\mathcal{H}}(w_t)\|^2\big]$$

$$= 3\kappa H^2 \gamma_t^2 G^2 [1 + 8L^2 H(H-1)\gamma_t^2] + 24\kappa L^2 H^3 (H-1)\gamma_t^4 \|\nabla \ell_{\mathcal{H}}(w_t)\|^2,$$

where (a) uses Eqs. (2) and (39), (b) uses Assumptions 2-3, (c) uses Assumption 2 and defines $V_{t,k} \overset{\text{def}}{=} \sum_{h=0}^{H-1} \|w_{t,h}^{(k)} - w_t\|^2$, (d) uses Eq. (37).

Then we prove Eq. (41) as follows.

$$\|\Delta_t + H\gamma_t \nabla \ell_{\mathcal{H}}(w_t)\|^2$$

$$\overset{(a)}{=} \left\| \frac{H\gamma_t}{|\mathcal{H}|} \sum_{k \in \mathcal{H}} \nabla \ell_k(w_t) - \frac{\gamma_t}{|\mathcal{H}|} \sum_{k \in \mathcal{H}} \sum_{h=0}^{H-1} \nabla \ell_k(w_{t,h}^{(k)}) \right\|^2$$

$$= H^2 \gamma_t^2 \left\| \frac{1}{H|\mathcal{H}|} \sum_{k \in \mathcal{H}} \sum_{h=0}^{H-1} [\nabla \ell_k(w_t) - \nabla \ell_k(w_{t,h}^{(k)})] \right\|^2$$

$$\leq H^2 \gamma_t^2 \cdot \frac{1}{H|\mathcal{H}|} \sum_{k \in \mathcal{H}} \sum_{h=0}^{H-1} \|\nabla \ell_k(w_t) - \nabla \ell_k(w_{t,h}^{(k)})\|^2$$

$$\overset{(b)}{\leq} \frac{L^2 H \gamma_t^2}{|\mathcal{H}|} \sum_{k \in \mathcal{H}} \sum_{h=0}^{H-1} \|w_{t,h}^{(k)} - w_t\|^2$$

$$\overset{(c)}{=} \frac{L^2 H \gamma_t^2}{|\mathcal{H}|} \sum_{k \in \mathcal{H}} V_{t,k}$$

$$\overset{(d)}{\leq} 4L^2 \gamma_t^4 H^3 (H-1) \big[G^2 + \|\nabla \ell_{\mathcal{H}}(w_t)\|^2\big],$$

where (a) uses Eqs. (1) and (39), (b) uses Assumption 2, (c) defines $V_{t,k} \overset{\text{def}}{=} \sum_{h=0}^{H-1} \|w_{t,h}^{(k)} - w_t\|^2$, and (d) uses Eq. (37). $\qquad\square$

**Lemma 5.** *For any* $0 \leq x < 1$, *we have*

$$\log(1-x) \leq -x \tag{42}$$

*Proof.* Denote the function $g(x) = \log(1-x)$, which has derivatives $g'(x) = (x-1)^{-1}$ and $g''(x) = -(x-1)^{-2}$. Then based on the Taylor's theorem, there exists $\theta \in [0,1]$ such that

$$\log(1-x) = g(x) = g(0) + g'(0)x + \frac{1}{2}g''(\theta x)x^2 = -x - \frac{x^2}{2(1-\theta x)^2} \leq -x.$$

$\qquad\square$

### E.2 Remaining Proof of Theorem 5

Using $L$-smoothness of $\ell_{\mathcal{H}}$, we have

$$\ell_{\mathcal{H}}(w_{t+1})$$

$$\leq \ell_{\mathcal{H}}(w_t) + \langle \nabla \ell_{\mathcal{H}}(w_t), w_{t+1} - w_t \rangle + \frac{L}{2} \|w_{t+1} - w_t\|^2$$

$$\leq \ell_{\mathcal{H}}(w_t) + \langle \nabla \ell_{\mathcal{H}}(w_t), w_{t+1} - w_t - \Delta_t \rangle + \langle \nabla \ell_{\mathcal{H}}(w_t), \Delta_t \rangle + L\|w_{t+1} - w_t - \Delta_t\|^2 + L\|\Delta_t\|^2$$

$$\overset{(a)}{\leq} \ell_{\mathcal{H}}(w_t) + \frac{H\gamma_t}{4}\|\nabla \ell_{\mathcal{H}}(w_t)\|^2 + \frac{1}{H\gamma_t}\|w_{t+1} - w_t - \Delta_t\|^2 + \frac{1}{2H\gamma_t}\|\Delta_t + H\gamma_t \nabla \ell_{\mathcal{H}}(w_t)\|^2$$

$$- \frac{H\gamma_t}{2}\|\nabla \ell_{\mathcal{H}}(w_t)\|^2 - \frac{1}{2H\gamma_t}\|\Delta_t\|^2 + L\|w_{t+1} - w_t - \Delta_t\|^2 + L\|\Delta_t\|^2$$

$$\overset{(b)}{\leq} \ell_{\mathcal{H}}(w_t) - \frac{H\gamma_t}{4}\|\nabla \ell_{\mathcal{H}}(w_t)\|^2 - \left(\frac{1}{2H\gamma_t} - L\right)\|\Delta_t\|^2$$

$$+ \frac{1}{2H\gamma_t} \cdot 4L^2 \gamma_t^4 H^3 (H-1)\big[G^2 + \|\nabla \ell_{\mathcal{H}}(w_t)\|^2\big]$$

$$+ \left(L + \frac{1}{H\gamma_t}\right)\big\{3\kappa H^2 \gamma_t^2 G^2 [1 + 8L^2 H(H-1)\gamma_t^2] + 24\kappa L^2 H^3 (H-1)\gamma_t^4 \|\nabla \ell_{\mathcal{H}}(w_t)\|^2\big\}$$

$$\overset{(c)}{\leq} \ell_{\mathcal{H}}(w_t) - \frac{H\gamma_t}{4}\big[1 - 8L^2 \gamma_t^2 H(H-1) - 96\kappa L^2 H(H-1)\gamma_t^2\big]\|\nabla \ell_{\mathcal{H}}(w_t)\|^2$$

$$+ G^2 H\gamma_t \{2L^2 \gamma_t^2 H(H-1) + 3\kappa(1 + LH\gamma_t)[1 + 8L^2 H(H-1)\gamma_t^2]\}, \tag{43}$$

where (a) uses $\langle u, v \rangle \leq \frac{H\gamma_t}{4}\|u\|^2 + \frac{1}{H\gamma_t}\|v\|^2$ for $u = \nabla \ell_{\mathcal{H}}(w_t)$ and $v = w_{t+1} - w_t - \Delta_t$, as well as $\langle \nabla \ell_{\mathcal{H}}(w_t), \Delta_t \rangle = \langle u, v \rangle = \frac{1}{2}(\|u + v\|^2 - \|u\|^2 - \|v\|^2)$ for $u = \sqrt{H\gamma_t}\nabla \ell_{\mathcal{H}}(w_t)$ and $v = \frac{\Delta_t}{\sqrt{H\gamma_t}}$, (b) uses Eqs. (40)-(41), (c) uses $\gamma_t \leq \frac{1}{2LH}$.

Select constant hyperparameters $\gamma_t = \gamma$ such that

$$L^2 \gamma^2 H(H-1) \leq \min\left(\frac{1}{32}, \frac{1}{384\kappa}\right), \gamma \leq \frac{1}{2LH}. \tag{44}$$

Then Eq. (43) simplifies into

$$\ell_{\mathcal{H}}(w_{t+1}) \leq \ell_{\mathcal{H}}(w_t) - \frac{H\gamma}{16}\|\nabla \ell_{\mathcal{H}}(w_t)\|^2 + G^2 H\gamma\Big[2L^2 \gamma^2 H(H-1) + 3\kappa\Big(1 + \frac{1}{2}\Big)\Big(1 + \frac{1}{4}\Big)\Big]. \tag{45}$$

Telescoping Eq. (45) above over $t = 0, 1, \ldots, T-1$, we have

$$\ell^* \leq \ell_{\mathcal{H}}(w_T) \leq \ell_{\mathcal{H}}(w_0) - \frac{H\gamma}{16}\sum_{t=0}^{T-1}\|\nabla \ell_{\mathcal{H}}(w_t)\|^2 + TG^2 H\gamma\Big[2L^2 \gamma^2 H(H-1) + \frac{45\kappa}{8}\Big].$$

Rearranging the inequality above, we prove the convergence rate (14) as follows.

$$\frac{1}{T}\sum_{t=0}^{T-1}\|\nabla \ell_{\mathcal{H}}(w_t)\|^2 \leq \frac{16}{TH\gamma}[\ell_{\mathcal{H}}(w_0) - \ell^*] + G^2[32L^2 \gamma^2 H(H-1) + 90\kappa]$$

$$\leq \frac{16cLH}{T^{2/3}}[\ell_{\mathcal{H}}(w_0) - \ell^*] + G^2\Big[\frac{32(H-1)}{c'^2 HT^{2/3}} + 90\kappa\Big]$$

$$\leq \frac{16cLH[\ell_{\mathcal{H}}(w_0) - \ell^*] + G^2}{T^{2/3}} + 90\kappa G^2,$$

where we select the stepsize $\gamma = \frac{1}{c'LHT^{1/3}}$ with $c' = \max(4\sqrt{2}, \sqrt{384\kappa})$ which satisfies the conditions in Eq. (44).

Furthermore, suppose Assumption 4 holds, that is,

$$\|\nabla \ell_{\mathcal{H}}(w)\|^2 \geq 2\mu(\ell_{\mathcal{H}}(w) - \ell^*).$$

Substituting the inequality above into Eq. (45), we have

$$\ell_{\mathcal{H}}(w_{t+1}) - \ell^* \leq \Big(1 - \frac{H\gamma\mu}{8}\Big)[\ell_{\mathcal{H}}(w_t) - \ell^*] + G^2 H\gamma\Big[2L^2 \gamma^2 H(H-1) + \frac{45\kappa}{8}\Big].$$

Iterating the inequality above over $t = 0, 1, \ldots, T-1$, we can prove Eq. (15) as follows.

$$\ell_{\mathcal{H}}(w_T) - \ell^* \leq \Big(1 - \frac{H\gamma\mu}{8}\Big)^T[\ell_{\mathcal{H}}(w_0) - \ell^*] + \frac{G^2}{\mu}\Big[16L^2 \gamma^2 H(H-1) + 45\kappa\Big]$$

Table 4: Architecture of ResNet-20 with Group Normalization (GN) for CIFAR-10

| Stage | Output Size | Layers |
|-------|-------------|--------|
| Input | $32 \times 32$ | $3 \times 3$ conv, 16 filters, stride 1 |
| Stage 1 | $32 \times 32$ | $3 \times$ [ $3 \times 3$ conv, 16 filters + GN($G = 4$) + ReLU ] |
| Stage 2 | $16 \times 16$ | $3 \times$ [ $3 \times 3$ conv, 32 filters + GN($G = 8$) + ReLU ] |
| Stage 3 | $8 \times 8$ | $3 \times$ [ $3 \times 3$ conv, 64 filters + GN($G = 8$) + ReLU ] |
| Output | $1 \times 1$ | Global average pooling, FC-10, softmax |

$$\overset{(a)}{\le} \exp\left[T \log\left(1 - \frac{\mu}{8c'LT^{1-\beta}}\right)\right][\ell_{\mathcal{H}}(w_0) - \ell^*] + \frac{16G^2}{\mu c'^2 T^{2-2\beta}} + \frac{45\kappa G^2}{\mu}$$

$$\overset{(b)}{\le} \exp\left[T\left(-\frac{\mu}{8c'LT^{1-\beta}}\right)\right][\ell_{\mathcal{H}}(w_0) - \ell^*] + \frac{G^2}{2\mu T^{2-2\beta}} + \frac{45\kappa G^2}{\mu}$$

$$\le \exp\left(-\frac{\mu T^{\beta}}{8c'L}\right)[\ell_{\mathcal{H}}(w_0) - \ell^*] + \frac{G^2}{2\mu T^{2-2\beta}} + \frac{45\kappa G^2}{\mu},$$

where (a) uses the stepsize $\gamma = \frac{1}{c'LHT^{1-\beta}}$, (b) uses Lemma 5 and $c' \ge 4\sqrt{2}$.

# F  EXPERIMENT DETAILS

## F.1  DIRICHLET-BASED PARTITION STRATEGIES

We adopt the Dirichlet-based partitioning scheme (Wang et al., 2020b) to simulate heterogeneous client data distributions for CIFAR-10. Specifically, we partition each dataset across clients using a Dirichlet-based distribution over class labels. In this setting, both the number of data points and the class proportions are imbalanced across clients. Specifically, we simulate a heterogeneous partition into $n$ clients by drawing class proportions from a Dirichlet distribution:

$$(p_{1,k}, p_{2,k}, \ldots, p_{10,k}) \sim \text{Dir}(\alpha_1, \alpha_2, \ldots, \alpha_{10}),$$

where $p_{c,k}$ denotes the proportion of training instances of class $c \in \{1, 2, \ldots, 10\}$ assigned to client $k$. The distribution $\text{Dir}_{10}(\cdot)$ is the 10-dimensional Dirichlet distribution. We set $\alpha_1 = \alpha_2 = \cdots = \alpha_K = \alpha$ to induce heterogeneity.

As a property of the Dirichlet distribution, when $\alpha_k < 1$, the sampled class proportions tend to concentrate near the corners and edges of the probability simplex, so that clients receive data from only a few dominant classes, thereby simulating severe label imbalance and statistical heterogeneity—common characteristics of practical federated learning environments.

Based on the sampled proportions $\{p_{c,k}\}$, we allocate the training data to each client accordingly. For evaluation, we use the original test set from each dataset as a global test set to ensure a fair comparison across all methods.

The number of training samples per client is the same across 16 clients. Since CIFAR-10 has 50,000 training examples, we assign each client $50,000/16 = 3,125$ training samples.

## F.2  MODEL DETAILS

Table 4 presents the detailed architecture of ResNet-20 with group normalization.

## F.3  TRAIN SCHEMES

**Preprocess of CIFAR-10:**  For preprocessing the images in the CIFAR-10 datasets, we follow the standard data augmentation and normalization procedures. Specifically, we apply random cropping and horizontal flipping for data augmentation. For normalization, each color channel is standardized

by subtracting the mean and dividing by the standard deviation of that channel. This ensures that the input images have zero mean and unit variance per channel, which is a common practice to stabilize and accelerate training.

**Local Update:** We perform one epoch of local updates per client when $\alpha = 0.1$, and two epochs of local updates per client when $\alpha = 1, 10$. Note that each epoch consists of multiple mini-batch SGD iterations, where each SGD step uses a batch size of 64. Therefore, we run FedRo with

$$H = \left\lfloor \frac{1 \times 3125}{64} \right\rfloor = 49$$

local SGD steps per round for $\alpha = 0.1$, and with

$$H = \left\lfloor \frac{2 \times 3125}{64} \right\rfloor = 98$$

local SGD steps per round for $\alpha = 1, 10$.

**Weight Decay:** We use L2 weight decay, i.e., and apply SGD to the cross-entropy loss with L2 regularizer $\frac{\mu}{2}\|w\|_2^2$ of the model parameter $w \in \mathbb{R}^d$. In our experiments, we set $\mu = 5 \times 10^{-4}$.

**Learning Rate Schedule:** We use a step-wise diminishing learning rate schedule as follows.

$$\gamma_t = \begin{cases} \gamma_0, & 0 \leq t < \frac{T}{2}, \\ 0.1\,\gamma_0, & \frac{T}{2} \leq t < \frac{3T}{4}, \\ 0.01\,\gamma_0, & \frac{3T}{4} \leq t \leq T, \end{cases} \tag{46}$$

where $T$ denotes the total number of communication rounds, and $\gamma_0$ is fine-tuned from $\{0.5, 0.2, 0.1, 0.05\}$.

# G  ADDITIONAL EXPERIMENTS

In this section, we conduct experiments with two composite aggregators ($\hat{f}$-CWTM$\circ\hat{f}$-NNM and $\hat{f}$-Krum$\circ\hat{f}$-NNM), as well as two additional attack methods that are different from the Gaussian attack, to verify our theoretical conclusions under different aggregators and attack types.

First, we define the two attack models considered in this section: **label-flip** and **sign-flip** attacks. In the label-flip attack, a Byzantine client corrupts its local dataset by replacing each ground-truth label $y$ with an incorrect label $\tilde{y}$. For multi-class classification, $\tilde{y}$ is chosen uniformly from the remaining classes. The client then performs local training on this corrupted dataset and sends the resulting update to the server. In the sign-flip attack, a Byzantine client keeps the original data but reverses the direction of its local update: if a benign client would upload an update $\Delta w_i$, a sign-flipping client instead uploads $-\omega \Delta w_i$, where $\omega \geq 1$ denotes the attack strength. These two attacks respectively model data-level corruption (label flip) and update-level corruption (sign flip), and are standard for testing the robustness of federated aggregation rules.

We follow a similar setting as in our aforementioned experiments. The only difference is that we set the attack strength to $\omega = 1$ and retune the hyperparameters.

Table 5 reports the top-1 accuracies of FedRo for each pair ($\hat{f}, \alpha$) under the two robust aggregators $\hat{f}$-CWTM$\circ\hat{f}$-NNM and $\hat{f}$-Krum$\circ\hat{f}$-NNM. When $\hat{f} < f = 4$(underestimation), both aggregators fail to effectively filter out Byzantine updates, and the accuracies remain close to random guessing ($\approx 10\%$) across all heterogeneity levels $\alpha$, which fits the divergence result in Theorem 3. When $\hat{f} \geq f$ (non-underestimation), the accuracies generally decrease as $\hat{f}$ increases beyond $f$. Moreover, this accuracy degradation becomes slower as $\alpha$ increases (i.e., as the data heterogeneity $G^2$ decreases): These results fit Theorems 4 and 5 which indicate that as $\hat{f}$ increases from $f$, the tight convergence lower bound $\frac{\hat{f}G^2}{n-f-\hat{f}}$ increases at a rate proportional to the heterogeneity $G^2$.

Throughout all configurations, $\hat{f}$-Krum$\circ\hat{f}$-NNM consistently yields slightly higher accuracies than $\hat{f}$-CWTM$\circ\hat{f}$-NNM and CWTM, but both exhibit the same qualitative trend, empirically confirming our theoretical results.

Table 5: Top-1 accuracies of the FedRo algorithm with two robust aggregators ($\hat{f}$-CWTM$\circ\hat{f}$-NNM and $\hat{f}$-Krum$\circ\hat{f}$-NNM) on CIFAR-10. A smaller $\alpha$ corresponds to higher heterogeneity $G^2$. When $\hat{f} > f$, the drops in accuracy relative to the corresponding accuracy at $\hat{f} = f$ (bolded) are shown in parentheses.

| $\hat{f}$ | $\hat{f}$-CWTM$\circ\hat{f}$-NNM | | | $\hat{f}$-Krum$\circ\hat{f}$-NNM | | |
|---|---|---|---|---|---|---|
| | $\alpha = 0.1$ | $\alpha = 1.0$ | $\alpha = 10.0$ | $\alpha = 0.1$ | $\alpha = 1.0$ | $\alpha = 10.0$ |
| 0 | 10.03 | 10.06 | 10.24 | 9.91 | 9.89 | 10.03 |
| 1 | 10.13 | 10.03 | 10.00 | 9.88 | 9.74 | 9.99 |
| 2 | 10.07 | 10.05 | 9.94 | 10.43 | 10.08 | 10.22 |
| 3 | 10.02 | 10.25 | 10.03 | 9.81 | 9.84 | 9.94 |
| 4 | **58.57** | **73.13** | **75.83** | **61.12** | **74.47** | **78.12** |
| 5 | 53.50 | 72.80 | 75.08 | 57.23 | 73.58 | 77.96 |
| 6 | 52.02 | 71.89 | 73.93 | 54.67 | 73.48 | 76.62 |
| 7 | 45.25 | 71.80 | 73.87 | 45.76 | 73.12 | 74.82 |

Table 6: Top-1 accuracies of the FedRo algorithm with $\hat{f}$-CWTM under two attacks (label-flip and sign-flip) on CIFAR-10. A smaller $\alpha$ corresponds to higher heterogeneity $G^2$. When $\hat{f} > f$, the drops in accuracy relative to the corresponding accuracy at $\hat{f} = f$ (bolded) are shown in parentheses.

| $\hat{f}$ | label-flip | | | sign-flip | | |
|---|---|---|---|---|---|---|
| | $\alpha = 0.1$ | $\alpha = 1.0$ | $\alpha = 10.0$ | $\alpha = 0.1$ | $\alpha = 1.0$ | $\alpha = 10.0$ |
| 0 | 29.10 | 56.32 | 65.28 | 10.00 | 13.85 | 20.61 |
| 1 | 31.76 | 57.82 | 67.04 | 18.88 | 38.55 | 34.47 |
| 2 | 39.61 | 68.90 | 69.36 | 27.40 | 45.00 | 53.92 |
| 3 | 40.50 | 68.62 | 69.07 | 33.90 | 47.72 | 54.94 |
| 4 | **55.46** | **72.41** | **76.48** | **50.84** | **68.26** | **73.42** |
| 5 | 53.47 | 71.99 | 75.65 | 47.05 | 62.92 | 70.22 |
| 6 | 52.80 | 71.53 | 75.28 | 45.28 | 59.86 | 67.37 |
| 7 | 48.10 | 70.01 | 75.07 | 43.36 | 58.92 | 66.84 |

Table 6 reports the top-1 accuracies of FedRo for each pair $(\hat{f}, \alpha)$ under the label-flip and sign-flip attacks. When $\hat{f} < f = 4$ (underestimation), under both attacks the algorithm fails to effectively filter out Byzantine updates, and the accuracies remain lower than those at $\hat{f} = 4$ across all heterogeneity levels $\alpha$. When $\hat{f} \geq f$ (non-underestimation), the results are similar to those of the previous experiments.

Note that, in the underestimation regime, the accuracies do not stay close to random guessing and still achieve non-trivial performance. This is because the attack strength is reduced to 1 and these attack types are weaker than the Gaussian attack.

## H  EXTENSION TO STOCHASTIC SETTING

Our convergence results can be extended to the stochastic federated learning problem. The objective is the same as Eq. (1) as repeated below.

$$\min_{w \in \mathbb{R}^d} \left\{ \ell_{\mathcal{H}}(w) \stackrel{\text{def}}{=} \frac{1}{|\mathcal{H}|} \sum_{k \in \mathcal{H}} \ell_k(w) \right\}. \tag{47}$$

The difference is that for stochastic problem, the loss function of each client $k$ can be expressed as the following stochastic form.

$$\ell_k(w) = \mathbb{E}_{\xi \sim \mathcal{P}_k}[\ell_k(w; \xi)], \tag{48}$$

where $\ell_k(w; \xi)$ is the loss function for a sample $\xi$ obtained from the data distribution $\mathcal{P}_k$ for the client $k$. As $\mathcal{P}_k$ usually involves large data, it becomes costly for every client $k$ to compute its full gradient

$\nabla\ell_k$. Instead, $\nabla\ell_k(w)$ can be approximated by the following stochastic gradient using the batch of i.i.d. samples $\mathcal{B} = \{\xi_i\}_{i=1}^{|\mathcal{B}|}$ from $\mathcal{P}_k$.

$$\nabla\ell_k(w;\mathcal{B}) = \frac{1}{|\mathcal{B}|}\sum_{\xi\in\mathcal{B}}\nabla\ell_k(w;\xi). \tag{49}$$

To adjust the FedRo algorithm (Algorithm 1) to the stochastic problem, we can replace the full gradient $\nabla\ell_k$ in Eq. (2) with the stochastic gradient $\nabla\ell_{k,\mathcal{B}_{k,t}}$ in Eq. (50), and obtain the StocFedRo algorithm (Algorithm 2) as follows.

---

**Algorithm 2** StocFedRo: Stochastic Federated Robust Averaging Algorithm

---

**Input:** The set of honest clients $\mathcal{H} \subset [n]$ with size $|\mathcal{H}| = n - f$ $(0 \le f < \frac{n}{2})$, number of rounds $T$, number of local parameter updates $H$, initial parameter $w_0 \in \mathbb{R}^d$, stepsize $\gamma_t$, batchsize $B_t$, aggregator $\mathcal{A}$.

**for** communication rounds $t = 0, 1, \ldots, T - 1$ **do**
    The server sends $w_t$ to every clients.
    **for** all clients $k \in [n]$ **in parallel do**
        **if** $k \in \mathcal{H}$ (honest client) **then**
            Initialize $w_{t,0}^{(k)} = w_t$.
            **for** $h = 0, \ldots, H - 1$ **do**
                Obtain $B_t$ i.i.d. samples $\mathcal{B}_{t,h}^{(k)}$ from $\mathcal{P}_k$, compute the stochastic gradient $\nabla\ell_k(w_{t,h}^{(k)};\mathcal{B}_{t,h}^{(k)})$ by Eq. (49), and then perform the following local parameter update:

$$w_{t,h+1}^{(k)} = w_{t,h}^{(k)} - \gamma_t\nabla\ell_k(w_{t,h}^{(k)};\mathcal{B}_{t,h}^{(k)}) \tag{50}$$

            **end**
            Upload $w_t^{(k)} = w_{t,H}^{(k)}$ to the server.
        **else**
            The Byzantine client $k$ uploads arbitrary $w_t^{(k)} \in \mathbb{R}^d$ to the server.
        **end**
    **end**
    The server updates the global parameter using aggregator $\mathcal{A}$.

$$w_{t+1} = w_t + \mathcal{A}(\{w_t^{(k)} - w_t\}_{k=1}^n). \tag{51}$$

**end**
**Output:** Select a parameter from $\{w_t\}_{t=0}^{T-1}$ uniformly at random.

---

To analyze the convergence of the StocFedRo Algorithm, we use the following assumption that is popular for stochastic federated learning problem (Huo et al., 2020; Li et al., 2020; Beikmohammadi et al., 2024).

**Assumption 5.** *For any $w \in \mathbb{R}^d$ and honest client $k \in \mathcal{H}$, the individual gradient $\nabla\ell_k(w;\xi)$ has variance bounded by a finite constant $\sigma^2 > 0$ as follows.*

$$\mathbb{E}_{\xi\sim\mathcal{P}_k}\|\nabla\ell_k(w;\xi) - \nabla\ell_k(w)\|^2 \le \sigma^2. \tag{52}$$

Then we can extend the convergence results (Theorems 3-5) to the stochastic problem respectively as the following Theorems 6-8.

**Theorem 6.** *For any $0 \le \hat{f} < f < \frac{n}{2}$, there exist an $(\hat{f}, \hat{\kappa})$-robust aggregator $\mathcal{A}$, a set $\mathcal{H} \subset [n]$ with size $|\mathcal{H}| = n - f$, Byzantine clients' strategies and stochastic loss functions $\{\ell_{i,\xi}\}_{i\in\mathcal{H}}$ satisfying Assumptions 1-5 such that when implementing Algorithm 2 with the aggregator $\mathcal{A}$, any initialization $w_0$, any constant stepsize $\gamma_t = \gamma > 0$ and any constant batchsize $B_t = B$, the generated sequence $w_t$ diverges as follows as $T \to +\infty$.*

$$\frac{1}{T}\sum_{t=0}^{T-1}\|\nabla\ell_{\mathcal{H}}(w_t)\|^2 \to +\infty \tag{53}$$

$$\ell_{\mathcal{H}}(w_T) - \ell^* \to +\infty \tag{54}$$

**Theorem 7** (Lower bound (Stochastic)). *For any $0 \leq f \leq \hat{f} < \frac{n}{2}$ and any $(\hat{f}, \hat{\kappa})$-robust aggregator $\mathcal{A}$, there exist $\mathcal{H} \subset [n]$ with size $|\mathcal{H}| = n - f$, Byzantine clients' strategies and stochastic loss functions $\{\ell_{i,\xi}\}_{i \in \mathcal{H}}$ satisfying Assumptions 1-5 such that for any initialization $w_0$ and constant stepsize $\gamma_t = \gamma > 0$, the sequence $w_t$ generated from Algorithm 2 with aggregator $\mathcal{A}$ either does not change over iteration (i.e., $w_t \equiv w_0$) or satisfies the following convergence lower bounds.*

$$\limsup_{T \to \infty} \frac{1}{T} \sum_{t=0}^{T-1} \|\nabla \ell_{\mathcal{H}}(w_t)\|^2 \geq \frac{\hat{f}G^2}{n - f - \hat{f}} \tag{55}$$

$$\limsup_{T \to \infty} \ell_{\mathcal{H}}(w_T) - \ell^* \geq \frac{\hat{f}G^2}{2\mu(n - f - \hat{f})} \tag{56}$$

**Theorem 8** (Upper bound (Stochastic)). *Suppose Assumptions 1-3 and 5 hold. Apply Algorithm 2 with an $(f, \kappa)$-aggregator and stepsize $\gamma = \frac{1}{\tilde{c}LHT^{1/3}}$ ($\tilde{c} = \max(4\sqrt{2}, 12\sqrt{2\kappa})$) to the stochastic federated optimization problem (47) with $f$ Byzantine clients. The algorithm output $\{w_t\}_{t=0}^{T-1}$ satisfies the following convergence rate.*

$$\frac{1}{T} \sum_{t=0}^{T-1} \|\nabla \ell_{\mathcal{H}}(w_t)\|^2 \leq \frac{16\tilde{c}LH[\ell_{\mathcal{H}}(w_0) - \ell^*] + G^2 + \sigma^2/(4B)}{T^{2/3}} + 216\kappa\Big(4G^2 + \frac{\sigma^2}{B}\Big). \tag{57}$$

*Furthermore, under Assumption 4 (i.e., $\ell_{\mathcal{H}}$ satisfies the PŁ condition), we can select stepsize $\gamma = \frac{1}{\tilde{c}LHT^{1-\beta}}$ for any $\beta \in (0, 1)$ which yields the following convergence rate.*

$$\ell_{\mathcal{H}}(w_T) - \ell^* \leq \exp\Big(-\frac{\mu T^\beta}{8\tilde{c}L}\Big)\mathbb{E}[\ell_{\mathcal{H}}(w_0) - \ell^*] + \Big(\frac{1}{16\mu T^{2-2\beta}} + \frac{54\kappa}{\mu}\Big)\Big(4G^2 + \frac{\sigma^2}{B}\Big). \tag{58}$$

**Remark:** These results for stochastic federated learning problem are very similar to those for deterministic problem. Specifically, in the underestimation case ($\hat{f} < f$), Theorem 6 indicates that divergence can also occur for stochastic problem. In the non-underestimation case ($\hat{f} \geq f$), Theorem 7 gives the same lower bounds $\frac{\hat{f}G^2}{n-f-\hat{f}} > 0$ (for gradient) and $\frac{\hat{f}G^2}{2\mu(n-f-\hat{f})} > 0$ (for function value) as Theorem 4 for deterministic problem. Theorem 8 provides the corresponding limiting upper bounds $\mathcal{O}\big[\kappa\big(G^2 + \frac{\sigma^2}{B}\big)\big]$ (for gradient) and $\mathcal{O}\big[\frac{\kappa}{\mu}\big(G^2 + \frac{\sigma^2}{B}\big)\big]$ (for function value) as $T \to +\infty$, which has the additional stochasticity-related constant $\frac{\sigma^2}{B}$ compared with the upper bounds given by Theorem 5 for deterministic problem. When selecting a good $(\hat{f}, \hat{\kappa})$-robust aggregator that is also $(f, \kappa)$-robust that attains the order-optimal upper bound $\kappa \leq \mathcal{O}\big(\frac{\hat{f}}{n-f-\hat{f}}\big)$ by Theorem 2, the upper bounds become $\mathcal{O}\big[\frac{\hat{f}}{n-f-\hat{f}}\big(G^2 + \frac{\sigma^2}{B}\big)\big]$ (for gradient) and $\mathcal{O}\big[\frac{\hat{f}}{\mu(n-f-\hat{f})}\big(G^2 + \frac{\sigma^2}{B}\big)\big]$ (for function value) respectively. In this case, both lower and upper bounds are proportional to $\frac{\hat{f}}{n-f-\hat{f}}$, which indicates that for stochastic problem, there is a similar trade-off in selecting $\hat{f}$: $\hat{f} \geq f$ is required to guarantee convergence, but the performance degrades as we increase $\hat{f}$ beyond $f$.

Next, we present the proof of Theorems 6-8 in the following subsections.

### H.1 PROOF OF THEOREMS 6 AND 7

Note that deterministic federated learning problem can be seen as a special case of the stochastic federated learning problem where $\ell_k(\cdot; \xi) \equiv \ell_k(\cdot)$ for all clients $k$ and samples $\xi$. In this case, the FedRo algorithm (Algorithm 1) for determinsitic problem is equivalent to the StocFedRo algorithm (Algorithm 2). Note that Theorem 3 can be seen as a deterministic special case of Theorem 6. Therefore, we can select loss functions $\ell_{k,\xi} \equiv \ell_k$ where $\ell_k$ meets the condition of Theorem 3, and the other hyperparameters are the same as those of Theorem 3, which proves Theorem 6. Theorem 7 can be proved in a similar way from its deterministic special case, Theorem 4.

First, we extend Lemma 3 to its stochastic version as follows.

**Lemma 6.** *Suppose $L^2\gamma_t^2 H^2 \leq \frac{1}{2}$. Then under Assumptions 3 and 5, the quantity $V_{t,k} \overset{\text{def}}{=} \sum_{h=0}^{H-1} \|w_{t,h}^{(k)} - w_t\|^2$ obtained from Algorithm 2 has the following upper bound.*

$$\sum_{k\in\mathcal{H}} \mathbb{E}[V_{t,k}] \leq \gamma_t^2 |\mathcal{H}| H^3 \Big[ 4\mathbb{E}[\|\nabla \ell_{\mathcal{H}}(w_t)\|^2] + 4G^2 + \frac{\sigma^2}{B_t} \Big] \tag{59}$$

*Proof.*

$$\mathbb{E}[V_{t,k}] \overset{\text{def}}{=} \sum_{h=0}^{H-1} \mathbb{E}[\|w_{t,h}^{(k)} - w_t\|^2]$$

$$\overset{(a)}{=} \sum_{h=0}^{H-1} \mathbb{E}\Big[\Big\| \sum_{h'=0}^{h-1} \gamma_t \nabla \ell_k(w_{t,h'}^{(k)}; \mathcal{B}_{t,h'}^{(k)}) \Big\|^2\Big]$$

$$\leq \gamma_t^2 \sum_{h=0}^{H-1}\sum_{h'=0}^{H-1} h\mathbb{E}[\|\nabla \ell_k(w_{t,h'}^{(k)}; \mathcal{B}_{t,h'}^{(k)})\|^2]$$

$$\overset{(b)}{\leq} \gamma_t^2 \sum_{h'=0}^{H-2}\sum_{h=h'+1}^{H-1} h\mathbb{E}\Big[\|\nabla \ell_k(w_{t,h'}^{(k)})\|^2 + \frac{\sigma^2}{B_t}\Big]$$

$$\overset{(c)}{\leq} \frac{\gamma_t^2 H(H-1)}{2} \sum_{h'=0}^{H-1} \mathbb{E}[\|\nabla \ell_k(w_{t,h'}^{(k)})\|^2] + \frac{\sigma^2\gamma_t^2}{B_t}\sum_{h'=0}^{H-2} \frac{H(H-1)}{2}$$

$$\leq \gamma_t^2 H(H-1) \sum_{h=0}^{H-1} \mathbb{E}\big[\|\nabla \ell_k(w_{t,h}^{(k)}) - \nabla \ell_k(w_t)\|^2 + \|\nabla \ell_k(w_t)\|^2\big] + \frac{H^3\sigma^2\gamma_t^2}{2B_t}$$

$$\leq L^2\gamma_t^2 H(H-1) \sum_{h=0}^{H-1} \mathbb{E}\big[\|w_{t,h}^{(k)} - w_t\|^2\big] + \gamma_t^2 H^2(H-1)\mathbb{E}\big[\|\nabla \ell_k(w_t)\|^2\big] + \frac{H^3\sigma^2\gamma_t^2}{2B_t}$$

$$\overset{(d)}{\leq} \frac{1}{2}\mathbb{E}[V_{t,k}] + \gamma_t^2 H^3 \mathbb{E}\Big[\|\nabla \ell_k(w_t)\|^2 + \frac{\sigma^2}{2B_t}\Big],$$

where (a) uses the local update rule (50), (b) swaps the two summations and uses Assumption 5, (c) uses $\sum_{h=h'+1}^{H-1} h \leq \sum_{h=1}^{H-1} h = \frac{H(H-1)}{2}$, and (d) uses $L^2\gamma_t^2 H^2 \leq \frac{1}{2}$. Rearranging the inequality above yields the following bound.

$$\mathbb{E}[V_{t,k}] \leq 2\gamma_t^2 H^3 \mathbb{E}\Big[\|\nabla \ell_k(w_t)\|^2 + \frac{\sigma^2}{2B_t}\Big].$$

Therefore, we can prove Eq. (59) by rearranging the inequality above over $k \in \mathcal{H}$ as follows.

$$\sum_{k\in\mathcal{H}} \mathbb{E}[V_{t,k}] \leq 2\gamma_t^2 H^3 \sum_{k\in\mathcal{H}} \mathbb{E}\Big[\|\nabla \ell_k(w_t)\|^2 + \frac{\sigma^2}{2B_t}\Big]$$

$$\leq 2\gamma_t^2 H^3 \sum_{k\in\mathcal{H}} \mathbb{E}\Big[2\|\nabla \ell_k(w_t) - \nabla \ell_{\mathcal{H}}(w_t)\|^2 + 2\|\nabla \ell_{\mathcal{H}}(w_t)\|^2 + \frac{\sigma^2}{2B_t}\Big]$$

$$\leq \gamma_t^2 |\mathcal{H}| H^3 \Big[ 4\mathbb{E}[\|\nabla \ell_{\mathcal{H}}(w_t)\|^2] + 4G^2 + \frac{\sigma^2}{B_t}\Big],$$

where the final $\leq$ uses Assumption 3.      $\square$

Then we extend Lemma 4 to its stochastic version as follows.

**Lemma 7.** *Define the following quantity obtained from Algorithm 2.*

$$\Delta_t \overset{\text{def}}{=} -\frac{\gamma_t}{|\mathcal{H}|} \sum_{k \in \mathcal{H}} \sum_{h=0}^{H-1} \nabla \ell_k(w_{t,h}^{(k)}). \tag{60}$$

*Suppose $L^2 \gamma_t^2 H^2 \leq \frac{1}{2}$ and the aggregator used in Algorithm 2 is $(f, \kappa)$-robust. Then under Assumptions 2, 3 and 5, $\Delta_t$ above satisfies the following bounds.*

$$\mathbb{E}[\|w_{t+1} - w_t - \Delta_t\|^2] \leq 9\kappa H^2 \gamma_t^2 \Big(G^2 + \frac{\sigma^2}{2B_t}\Big) + 12L^2 \kappa H^4 \gamma_t^4 \mathbb{E}[\|\nabla \ell_{\mathcal{H}}(w_t)\|^2], \tag{61}$$

$$\mathbb{E}[\|\Delta_t + H\gamma_t \nabla \ell_{\mathcal{H}}(w_t)\|^2] \leq L^2 H^4 \gamma_t^4 \Big[4\mathbb{E}[\|\nabla \ell_{\mathcal{H}}(w_t)\|^2] + 4G^2 + \frac{\sigma^2}{B_t}\Big], \tag{62}$$

*Proof.* Since $\mathcal{A}$ is an $(f, \kappa)$-robust aggregator ($f = n - |\mathcal{H}|$) by Definition 1, we can prove Eq. (61) as follows.

$$\mathbb{E}[\|w_{t+1} - w_t - \Delta_t\|^2]$$

$$= \mathbb{E}[\|\mathcal{A}(\{w_t^{(k)} - w_t\}_{k=1}^n) - \Delta_t\|^2]$$

$$\leq \frac{\kappa}{|\mathcal{H}|} \sum_{k \in \mathcal{H}} \mathbb{E}[\|w_t^{(k)} - w_t - \Delta_t\|^2]$$

$$\overset{(a)}{=} \frac{\kappa}{|\mathcal{H}|} \sum_{k \in \mathcal{H}} \mathbb{E}\Big[\Big\| - \gamma_t \sum_{h=0}^{H-1} \nabla \ell_k(w_{t,h}^{(k)}; \mathcal{B}_{t,h}^{(k)}) + \frac{\gamma_t}{|\mathcal{H}|} \sum_{k' \in \mathcal{H}} \sum_{h=0}^{H-1} \nabla \ell_{k'}(w_{t,h}^{(k')}) \Big\|^2\Big]$$

$$= \frac{\kappa \gamma_t^2}{|\mathcal{H}|} \sum_{k \in \mathcal{H}} \mathbb{E}\Big\{\Big\| \sum_{h=0}^{H-1} \Big[\nabla \ell_k(w_{t,h}^{(k)}; \mathcal{B}_{t,h}^{(k)}) - \frac{1}{|\mathcal{H}|} \sum_{k' \in \mathcal{H}} \nabla \ell_{k'}(w_{t,h}^{(k')})\Big] \Big\|^2\Big\}$$

$$\leq \frac{\kappa H \gamma_t^2}{|\mathcal{H}|} \sum_{k \in \mathcal{H}} \sum_{h=0}^{H-1} \mathbb{E}\Big[\Big\| \nabla \ell_k(w_{t,h}^{(k)}; \mathcal{B}_{t,h}^{(k)}) - \frac{1}{|\mathcal{H}|} \sum_{k' \in \mathcal{H}} \nabla \ell_{k'}(w_{t,h}^{(k')}) \Big\|^2\Big]$$

$$= \kappa H \gamma_t^2 \sum_{h=0}^{H-1} \frac{1}{|\mathcal{H}|} \sum_{k \in \mathcal{H}} \mathbb{E}\Big[\Big\| \nabla \ell_k(w_{t,h}^{(k)}; \mathcal{B}_{t,h}^{(k)}) - \frac{1}{|\mathcal{H}|} \sum_{k' \in \mathcal{H}} \nabla \ell_{k'}(w_{t,h}^{(k')}) \Big\|^2\Big]$$

$$\leq 3\kappa H \gamma_t^2 \sum_{h=0}^{H-1} \frac{1}{|\mathcal{H}|} \sum_{k \in \mathcal{H}} \mathbb{E}\Big[\Big\| \nabla \ell_k(w_t) - \frac{1}{|\mathcal{H}|} \sum_{k' \in \mathcal{H}} \nabla \ell_{k'}(w_t) \Big\|^2$$

$$+ \Big\| \nabla \ell_k(w_{t,h}^{(k)}; \mathcal{B}_{t,h}^{(k)}) - \nabla \ell_k(w_t) \Big\|^2 + \Big\| \frac{1}{|\mathcal{H}|} \sum_{k' \in \mathcal{H}} [\nabla \ell_{k'}(w_t) - \nabla \ell_{k'}(w_{t,h}^{(k')})] \Big\|^2\Big]$$

$$\overset{(b)}{\leq} 3\kappa H \gamma_t^2 \sum_{h=0}^{H-1} \frac{1}{|\mathcal{H}|} \sum_{k \in \mathcal{H}} \mathbb{E}\Big[\Big\| \nabla \ell_k(w_t) - \frac{1}{|\mathcal{H}|} \sum_{k' \in \mathcal{H}} \nabla \ell_{k'}(w_t) \Big\|^2\Big]$$

$$+ 3\kappa H \gamma_t^2 \sum_{h=0}^{H-1} \frac{1}{|\mathcal{H}|} \sum_{k \in \mathcal{H}} \frac{\sigma^2}{B_t} + 3\kappa H \gamma_t^2 \sum_{h=0}^{H-1} \frac{1}{|\mathcal{H}|} \sum_{k' \in \mathcal{H}} \mathbb{E}[\|\nabla \ell_{k'}(w_t) - \nabla \ell_{k'}(w_{t,h}^{(k')})\|^2]$$

$$\overset{(c)}{\leq} 3\kappa H^2 \gamma_t^2 \Big(G^2 + \frac{\sigma^2}{B_t}\Big) + \frac{3L^2 \kappa H \gamma_t^2}{|\mathcal{H}|} \sum_{h=0}^{H-1} \sum_{k \in \mathcal{H}} \mathbb{E}[\|w_{t,h}^{(k)} - w_t\|^2]$$

$$\overset{(d)}{=} 3\kappa H^2 \gamma_t^2 \Big(G^2 + \frac{\sigma^2}{B_t}\Big) + \frac{3L^2 \kappa H \gamma_t^2}{|\mathcal{H}|} \sum_{k \in \mathcal{H}} \mathbb{E}[V_{t,k}]$$

$$\overset{(e)}{\leq} 3\kappa H^2 \gamma_t^2 \Big(G^2 + \frac{\sigma^2}{B_t}\Big) + \frac{3L^2 \kappa H \gamma_t^2}{|\mathcal{H}|} \cdot \gamma_t^2 |\mathcal{H}| H^3 \Big[4\mathbb{E}[\|\nabla \ell_{\mathcal{H}}(w_t)\|^2] + 4G^2 + \frac{\sigma^2}{B_t}\Big]$$

$$\overset{(f)}{\leq} 3\kappa H^2 \gamma_t^2 \Big(G^2 + \frac{\sigma^2}{B_t}\Big) + \frac{3\kappa H^2 \gamma_t^2}{2} \Big(4G^2 + \frac{\sigma^2}{B_t}\Big) + 12L^2 \kappa H^4 \gamma_t^4 \mathbb{E}[\|\nabla \ell_{\mathcal{H}}(w_t)\|^2]$$

$$=9\kappa H^2\gamma_t^2\Big(G^2+\frac{\sigma^2}{2B_t}\Big)+12L^2\kappa H^4\gamma_t^4\mathbb{E}[\|\nabla\ell_{\mathcal{H}}(w_t)\|^2],$$

where (a) uses Eqs. (50) and (60), (b) uses Assumption 5 and applies Jensen's inequality to the convex function $\|\cdot\|^2$, (c) uses Assumptions 2 and 3, (d) defines $V_{t,k}\overset{\text{def}}{=}\sum_{h=0}^{H-1}\|w_{t,h}^{(k)}-w_t\|^2$, (e) uses Eq. (59), and (f) uses $L^2\gamma_t^2 H^2\le\frac{1}{2}$. Then we prove Eq. (62) as follows.

$$\mathbb{E}[\|\Delta_t+H\gamma_t\nabla\ell_{\mathcal{H}}(w_t)\|^2]$$

$$\overset{(a)}{=}\mathbb{E}\Big[\Big\|\frac{H\gamma_t}{|\mathcal{H}|}\sum_{k\in\mathcal{H}}\nabla\ell_k(w_t)-\frac{\gamma_t}{|\mathcal{H}|}\sum_{k\in\mathcal{H}}\sum_{h=0}^{H-1}\nabla\ell_k(w_{t,h}^{(k)})\Big\|^2\Big]$$

$$=H^2\gamma_t^2\mathbb{E}\Big[\Big\|\frac{1}{H|\mathcal{H}|}\sum_{k\in\mathcal{H}}\sum_{h=0}^{H-1}[\nabla\ell_k(w_t)-\nabla\ell_k(w_{t,h}^{(k)})]\Big\|^2\Big]$$

$$\le H^2\gamma_t^2\cdot\frac{1}{H|\mathcal{H}|}\sum_{k\in\mathcal{H}}\sum_{h=0}^{H-1}\mathbb{E}[\|\nabla\ell_k(w_t)-\nabla\ell_k(w_{t,h}^{(k)})\|^2]$$

$$\overset{(b)}{\le}\frac{L^2 H\gamma_t^2}{|\mathcal{H}|}\sum_{k\in\mathcal{H}}\sum_{h=0}^{H-1}\mathbb{E}[\|w_{t,h}^{(k)}-w_t\|^2]$$

$$\overset{(c)}{=}\frac{L^2 H\gamma_t^2}{|\mathcal{H}|}\sum_{k\in\mathcal{H}}\mathbb{E}[V_{t,k}]$$

$$\overset{(d)}{\le}L^2 H^4\gamma_t^4\Big[4\mathbb{E}[\|\nabla\ell_{\mathcal{H}}(w_t)\|^2]+4G^2+\frac{\sigma^2}{B_t}\Big],$$

where (a) uses Eqs. (47) and (60), (b) uses Assumption 2, (c) defines $V_{t,k}\overset{\text{def}}{=}\sum_{h=0}^{H-1}\|w_{t,h}^{(k)}-w_t\|^2$, and (d) uses Eq. (59). $\qquad\square$

## H.3   REMAINING PROOF OF THEOREM 8

Using $L$-smoothness of $\ell_{\mathcal{H}}$, we have

$$\mathbb{E}[\ell_{\mathcal{H}}(w_{t+1})]$$

$$\le\mathbb{E}\Big[\ell_{\mathcal{H}}(w_t)+\langle\nabla\ell_{\mathcal{H}}(w_t),w_{t+1}-w_t\rangle+\frac{L}{2}\|w_{t+1}-w_t\|^2\Big]$$

$$\le\mathbb{E}\Big[\ell_{\mathcal{H}}(w_t)+\langle\nabla\ell_{\mathcal{H}}(w_t),w_{t+1}-w_t-\Delta_t\rangle+\langle\nabla\ell_{\mathcal{H}}(w_t),\Delta_t\rangle+L\|w_{t+1}-w_t-\Delta_t\|^2+L\|\Delta_t\|^2\Big]$$

$$\overset{(a)}{\le}\mathbb{E}\Big[\ell_{\mathcal{H}}(w_t)+\frac{H\gamma_t}{4}\|\nabla\ell_{\mathcal{H}}(w_t)\|^2+\frac{1}{H\gamma_t}\|w_{t+1}-w_t-\Delta_t\|^2+\frac{1}{2H\gamma_t}\|\Delta_t+H\gamma_t\nabla\ell_{\mathcal{H}}(w_t)\|^2$$

$$-\frac{H\gamma_t}{2}\|\nabla\ell_{\mathcal{H}}(w_t)\|^2-\frac{1}{2H\gamma_t}\|\Delta_t\|^2+L\|w_{t+1}-w_t-\Delta_t\|^2+L\|\Delta_t\|^2\Big]$$

$$\overset{(b)}{\le}\mathbb{E}[\ell_{\mathcal{H}}(w_t)]-\frac{H\gamma_t}{4}\mathbb{E}[\|\nabla\ell_{\mathcal{H}}(w_t)\|^2]-\Big(\frac{1}{2H\gamma_t}-L\Big)\mathbb{E}[\|\Delta_t\|^2]$$

$$+\frac{1}{2H\gamma_t}\cdot L^2 H^4\gamma_t^4\Big[4\mathbb{E}[\|\nabla\ell_{\mathcal{H}}(w_t)\|^2]+4G^2+\frac{\sigma^2}{B_t}\Big]$$

$$+\Big(L+\frac{1}{H\gamma_t}\Big)\Big[9\kappa H^2\gamma_t^2\Big(G^2+\frac{\sigma^2}{2B_t}\Big)+12L^2\kappa H^4\gamma_t^4\mathbb{E}[\|\nabla\ell_{\mathcal{H}}(w_t)\|^2]\Big]$$

$$\overset{(c)}{\le}\mathbb{E}[\ell_{\mathcal{H}}(w_t)]-\frac{H\gamma_t}{4}\mathbb{E}[\|\nabla\ell_{\mathcal{H}}(w_t)\|^2]+L^2 H^3\gamma_t^3\Big[2\mathbb{E}[\|\nabla\ell_{\mathcal{H}}(w_t)\|^2]+2G^2+\frac{\sigma^2}{2B_t}\Big]$$

$$+\frac{3}{2H\gamma_t}\Big[9\kappa H^2\gamma_t^2\Big(G^2+\frac{\sigma^2}{2B_t}\Big)+12L^2\kappa H^4\gamma_t^4\mathbb{E}[\|\nabla\ell_{\mathcal{H}}(w_t)\|^2]\Big]$$

$$\le\mathbb{E}[\ell_{\mathcal{H}}(w_t)]-\frac{H\gamma_t}{4}(1-8L^2 H^2\gamma_t^2-72L^2\kappa H^2\gamma_t^2)\mathbb{E}[\|\nabla\ell_{\mathcal{H}}(w_t)\|^2]$$

$$+ L^2 H^3 \gamma_t^3 \Big( 2G^2 + \frac{\sigma^2}{2B_t} \Big) + \frac{27\kappa H \gamma_t}{2} \Big( G^2 + \frac{\sigma^2}{2B_t} \Big), \tag{63}$$

where (a) uses $\langle u, v \rangle \leq \frac{H\gamma_t}{4} \|u\|^2 + \frac{1}{H\gamma_t} \|v\|^2$ for $u = \nabla \ell_{\mathcal{H}}(w_t)$ and $v = w_{t+1} - w_t - \Delta_t$, as well as $\langle \nabla \ell_{\mathcal{H}}(w_t), \Delta_t \rangle = \langle u, v \rangle = \frac{1}{2}(\|u+v\|^2 - \|u\|^2 - \|v\|^2)$ for $u = \sqrt{H\gamma_t} \nabla \ell_{\mathcal{H}}(w_t)$ and $v = \frac{\Delta_t}{\sqrt{H\gamma_t}}$, (b) uses Eqs. (61)-(62), and (c) uses $\gamma_t \leq \frac{1}{2LH}$.

Select constant hyperparameters $\gamma_t = \gamma$ and $B_t = B$ such that

$$L^2 \gamma^2 H^2 \leq \min \Big( \frac{1}{32}, \frac{1}{288\kappa} \Big). \tag{64}$$

Then Eq. (63) simplifies into

$$\mathbb{E}[\ell_{\mathcal{H}}(w_{t+1})] \leq \mathbb{E}[\ell_{\mathcal{H}}(w_t)] - \frac{H\gamma}{16} \mathbb{E}[\|\nabla \ell_{\mathcal{H}}(w_t)\|^2] + L^2 H^3 \gamma^3 \Big( 2G^2 + \frac{\sigma^2}{2B} \Big)$$
$$+ \frac{27\kappa H \gamma}{2} \Big( G^2 + \frac{\sigma^2}{2B} \Big). \tag{65}$$

Telescoping Eq. (65) above over $t = 0, 1, \ldots, T-1$, we have

$$\ell^* \leq \mathbb{E}[\ell_{\mathcal{H}}(w_T)] \leq \mathbb{E}[\ell_{\mathcal{H}}(w_0)] - \frac{H\gamma}{16} \sum_{t=0}^{T-1} \mathbb{E}[\|\nabla \ell_{\mathcal{H}}(w_t)\|^2]$$
$$+ TL^2 H^3 \gamma^3 \Big( 2G^2 + \frac{\sigma^2}{2B} \Big) + \frac{27T\kappa H \gamma}{2} \Big( G^2 + \frac{\sigma^2}{2B} \Big).$$

Rearranging the inequality above, we prove the convergence rate (57) as follows.

$$\frac{1}{T} \sum_{t=0}^{T-1} \mathbb{E}[\|\nabla \ell_{\mathcal{H}}(w_t)\|^2]$$
$$\leq \frac{16}{TH\gamma} \mathbb{E}[\ell_{\mathcal{H}}(w_0) - \ell^*] + 16 L^2 H^2 \gamma^2 \Big( 2G^2 + \frac{\sigma^2}{2B} \Big) + 432\kappa \Big( G^2 + \frac{\sigma^2}{2B} \Big)$$
$$\leq \frac{16\tilde{c}LH}{T^{2/3}} [\ell_{\mathcal{H}}(w_0) - \ell^*] + \Big( 2G^2 + \frac{\sigma^2}{2B} \Big) \Big[ \frac{16}{\tilde{c}^2 T^{2/3}} + 432\kappa \Big]$$
$$\leq \frac{16\tilde{c}LH[\ell_{\mathcal{H}}(w_0) - \ell^*] + G^2 + \sigma^2/(4B)}{T^{2/3}} + 216\kappa \Big( 4G^2 + \frac{\sigma^2}{B} \Big),$$

where we select the stepsize $\gamma = \frac{1}{\tilde{c}LHT^{1/3}}$ with $\tilde{c} = \max(4\sqrt{2}, 2\sqrt{2\kappa})$ which satisfies the conditions in Eq. (64).

Furthermore, suppose Assumption 4 holds, that is,

$$\|\nabla \ell_{\mathcal{H}}(w)\|^2 \geq 2\mu(\ell_{\mathcal{H}}(w) - \ell^*).$$

Substituting the inequality above into Eq. (65), we have

$$\mathbb{E}[\ell_{\mathcal{H}}(w_{t+1}) - \ell^*] \leq \Big( 1 - \frac{H\gamma\mu}{8} \Big) \mathbb{E}[\ell_{\mathcal{H}}(w_t) - \ell^*] + \Big( L^2 H^3 \gamma^3 + \frac{27\kappa H \gamma}{2} \Big) \Big( 2G^2 + \frac{\sigma^2}{2B} \Big).$$

Iterating the inequality above over $t = 0, 1, \ldots, T-1$, we can prove Eq. (58) as follows.

$$\mathbb{E}[\ell_{\mathcal{H}}(w_T) - \ell^*]$$
$$\leq \Big( 1 - \frac{H\gamma\mu}{8} \Big)^T \mathbb{E}[\ell_{\mathcal{H}}(w_0) - \ell^*] + \frac{8}{H\gamma\mu} \Big( L^2 H^3 \gamma^3 + \frac{27\kappa H \gamma}{2} \Big) \Big( 2G^2 + \frac{\sigma^2}{2B} \Big)$$
$$\leq \exp \Big[ T \log \Big( 1 - \frac{\mu}{8\tilde{c}LT^{1-\beta}} \Big) \Big] \mathbb{E}[\ell_{\mathcal{H}}(w_0) - \ell^*] + \frac{2}{\mu} \Big( L^2 H^2 \gamma^2 + 27\kappa \Big) \Big( 4G^2 + \frac{\sigma^2}{B} \Big)$$
$$\stackrel{(a)}{\leq} \exp \Big[ T \Big( - \frac{\mu}{8\tilde{c}LT^{1-\beta}} \Big) \Big] \mathbb{E}[\ell_{\mathcal{H}}(w_0) - \ell^*] + \frac{2}{\mu} \Big( \frac{1}{32T^{2-2\beta}} + 27\kappa \Big) \Big( 4G^2 + \frac{\sigma^2}{B} \Big)$$
$$\leq \exp \Big( - \frac{\mu T^\beta}{8\tilde{c}L} \Big) \mathbb{E}[\ell_{\mathcal{H}}(w_0) - \ell^*] + \Big( \frac{1}{16\mu T^{2-2\beta}} + \frac{54\kappa}{\mu} \Big) \Big( 4G^2 + \frac{\sigma^2}{B} \Big),$$

where (a) uses the stepsize $\gamma = \frac{1}{\tilde{c}LHT^{1-\beta}} \leq \frac{1}{4\sqrt{2}LHT^{1-\beta}}$ and Lemma 5.