# OpenReview forum: "Trade-off in Estimating the Number of Byzantine Clients in Federated Learning"
_ICLR.cc/2026/Conference — Submitted to ICLR 2026_

### Official Review · Reviewer_teAR · 2025-10-17

**Soundness:** 4
**Presentation:** 3
**Contribution:** 2
**Rating:** 4
**Confidence:** 5

**Summary:**

The paper studies Byzantine-robust federated learning when the server only knows an estimate \(\hat f\) of the number of Byzantine clients (the true number is \(f\)).
Main takeaways: (i) if we underestimate  (\(\hat f < f\)), aggregation / training can become arbitrarily bad; (ii) if we do not underestimate (\(\hat f \ge f\)), the best-possible error (and convergence floor) scales like
\[
\kappa = \Theta\!\left(\frac{\hat f}{\,n - f - \hat f\,}\right),
\]
with matching lower/upper bounds and a simple construction that achieves this rate up to constants.

**Strengths:**

- Clear split between two regimes: underestimation can be catastrophic; otherwise there is a tight, quantified floor.
- Matching lower and upper bounds
- The dependence on both \(f\) and \(\hat f\) is explicit, which clarifies the cost of being conservative.

**Weaknesses:**

- The underestimation impossibility is not really new: it is closely tied to the **breakdown point** in robust statistics.
  If the true contamination $f/n$ exceeds the method's breakdown (here essentially $\hat f/n$), you can get unbounded error.
  So this part feels more like transferring a known concept, rather than introducing a new phenomenon.

- Novelty in the *order* of the bound is modest. Away from edges, it matches the trivial (previously known) bound up to constants.
  Precisely, for a constant $c \in (0, \tfrac{1}{2})$ such that
  $$
  \hat f \le \big(\tfrac{1}{2} - c\big)\,n
  \quad \text{and} \quad
  f + \hat f \le (1 - c)\,n,
  $$
  both denominators are $\Theta(n)$, and
  $$
  \frac{\hat f/(n - f - \hat f)}{\hat f/(n - 2\hat f)}
  \;=\; \frac{n - 2\hat f}{\,n - f - \hat f\,}
  \;=\; \frac{1}{\,1 + \frac{\hat f - f}{\,n - 2\hat f\,}\,}.
  $$
  This ratio stays bounded between two positive constants (and $<1$ when $\hat f > f$),
  so the two expressions are of the *same order*.

- They only separate near the **edges**:
  - $f + \hat f \to n$: the paper’s denominator $n - f - \hat f$ collapses $\Rightarrow$ larger floor than the trivial count.
  - $\hat f \to n/2$: the trivial denominator $n - 2\hat f$ collapses $\Rightarrow$ trivial bound blows up sooner.
  - **Underestimation** $\hat f < f$: the paper (consistent with breakdown) shows no finite guarantee, while the trivial bound misleadingly remains finite.

- Experiments could be broader; I appreciate the paper is from a theoretical nature so this is a minor issue for me.

**Questions:**

- Since the underestimation result follows the breakdown-point logic, what is the extra conceptual significance here?

see weaknesses

---

> ### Author Response · Authors · 2025-11-20
> **Authors' Response to Reviewer teAR**
>
> Thank you for your precious comments. We have replied to your questions below and uploaded the revised paper with the revision shown in red.
>
> Q1: The underestimation impossibility is not really new: it is closely tied to the breakdown point in robust statistics. If the true contamination
> $f/n$ exceeds the method's breakdown (here essentially $\hat{f}/n$), you can get unbounded error. So this part feels more like transferring a known concept, rather than introducing a new phenomenon. Since the underestimation result follows the breakdown-point logic, what is the extra conceptual significance here?
>
> A1: Our underestimation theory formally relates the notion of $(f,\kappa)$-robustness to breakdown point, and indicates that when using an $(\hat{f},\hat{\kappa})$-robust aggregator, the breakpoint for both aggregation and federated learning is strictly larger than $\hat{f}$. While this underestimation theory is not significantly novel, we keep it since it is complementary to our more novel non-underestimation theory to reach the whole conclusion of trade-off in selecting $\hat{f}$.
>
> Q2: The relationship between our bound and previous trivial bound.
>
> A2: The previous trivial bound $\frac{\hat{f}}{n-2\hat{f}}$ holds **only when** applying $(\hat{f},\hat{\kappa})$-robust aggregator to the problem with $f$ actual Byzantine clients with $\hat{f}=f$. When $\hat{f}>f$ (overestimation), the $(\hat{f},\hat{\kappa})$-robust aggregator is also $(f,\hat{\kappa})$-robust based on item 1 of our Theorem 2, so the previous trivial lower bound is $\frac{f}{n-2f}$ **NOT** $\frac{\hat{f}}{n-2\hat{f}}$. This previous trivial lower bound $\frac{f}{n-2f}$ is strictly less than our $\frac{\hat{f}}{n-f-\hat{f}}$ when $\hat{f}>f$.
>
> Q3: Experiments could be broader; I appreciate the paper is from a theoretical nature so this is a minor issue for me.
>
> A3: Thanks for your suggestion. In the Appendix G, we have added experiments on more aggregators and attacks, which indicate results that are consistent with our previous experimental results.

---

> > ### Comment · Reviewer_teAR · 2025-11-25
> >
> > Thank you for the response.
> >
> > I am not quite sure I got the response to one of the concerns raised. Do the authors agree that the order of the new bounds only differs from the previous bound in the edge cases, and otherwise the two have the same order as explained above?
> >
> > Thanks!

---

> ### Author Response · Authors · 2025-11-25
> **Comparing bounds**
>
> The previous bound lower bound $\frac{f}{n-2f}$ has strictly lower order than our $\frac{\hat{f}}{n-f-\hat{f}}$ under order-level overestimation $\hat{f}\gg f$ (i.e., $\hat{f}$ has higher order than that of $f$). This covers many cases  (**NOT only edge cases**), for example,
>
> (1) When $f=\mathcal{O}(1)$, while $\hat{f}=\mathcal{O}(n^a)$ ($0<a\le 1$) or $\hat{f}=\mathcal{O}(\log^a n)$ ($a>0$);
>
> (2) When $f=\mathcal{O}(\log^b n)$ ($b>0$), while $\hat{f}=\log^a n$ ($a>b$) or $\hat{f}=\mathcal{O}(n^a)$ ($0<a\le 1$);
>
> (3) When $f=\mathcal{O}(n^a)$ ($0<a<1$) while $\hat{f}=\mathcal{O}(n^b)$ ($a<b\le 1$).

---

### Official Review · Reviewer_dsqW · 2025-10-29

**Soundness:** 2
**Presentation:** 3
**Contribution:** 3
**Rating:** 4
**Confidence:** 5

**Summary:**

This paper investigates the effect of underestimation and overestimation of the number of Byzantine clients and highlights the importance of accurately estimating the Byzantine size. The authors theoretically prove and empirically validate that underestimation can lead to divergence under Byzantine attacks, while overestimation results in a trade-off: a larger estimated Byzantine size enhances robustness under Byzantine attacks but degrades performance when there are no or fewer Byzantine clients.

**Strengths:**

1. The investigation of the impact of under- and over-estimating the number of Byzantine clients in the context of Byzantine attacks is a relevant and important topic for the field of Byzantine-robust federated learning.

2. This paper is well-written, clear, and theoretically sound.

**Weaknesses:**

1. The theoretical results in Theorems 1 and 3 do not fully support the conclusion that underestimation leads to arbitrarily large aggregation and convergence errors, as they apply only to a specific robust aggregator, not all such aggregators. If a different robust aggregator is used, the effect of underestimation may not hold. This represents a notable limitation of the paper. Can the authors generalize the results in Theorems 1 and 3 to all robust aggregators to address this issue?

2. There is a discrepancy between the theoretical analysis and the empirical results, as the former is conducted in a deterministic setting, while the latter is based on a stochastic setting. This gap is significant, as stochastic noise can substantially affect the performance of FedRo and lead to different convergence properties. Can the authors extend the analysis to address this gap?

3. In Table 1, the robustness coefficient $\kappa$ for the combination of $\hat{f}$-Krum and $\hat{f}$-NNM matches the lower bound in order. Why does the authors not use that aggregator in experiments? The reviewer would like to see the results of using $\hat{f}$-Krum $\circ$ $\hat{f}$-NNM in experiments.

4. In Line 48, the authors claim that "Existing works typically require estimating the actual number $f$ or the fraction of the Byzantine clients to select the maximum number $\hat{f}$ that the aggregator can tolerate." To my knowledge, the tolerable maximum number of Byzantine clients (i.e., the breakdown point) for some robust aggregators is independent of the actual number $f$ or fraction of Byzantine clients, meaning that these terms do not need to be estimated in advance for such aggregators. For instance, $\frac{n}{2}$ for CWMed, $\frac{n-2}{2}$ for CWTM, and $\frac{n}{10}$ for centered clipping, with respect to the breakdown point. The authors should revise this statement to make it more precise and rigorous.

5. In line 237, the authors claim that "since $\kappa$ of the commonly used aggregators like GM, CWTM, CWMed, and Krum do not match the lower bound even in the simple special case of $f = \hat{f}$." This statement is incorrect. As noted in Remark 2 of Allouah et al. (2023), CWTM does match the lower bound in order. The reviewer suggests that the authors carefully verify this statement.

6. There are several typos in this paper. For instance, in line 225, $\frac{\hat{f}}{n - 2f}$ should be $\frac{\hat{f}}{n - 2\hat{f}}$, and in line 226, $\frac{\hat{f}}{n - 2\hat{f}}$ should be $\frac{\hat{f}}{n - \hat{f}}$. The authors are advised to carefully review the paper to address these and prevent any similar issues.

**Questions:**

My detailed questions are outlined in the section above; please refer to them. If the authors can fully address my concerns, I will consider adjusting my score accordingly.

---

> ### Author Response · Authors · 2025-11-20
> **Authors' Response to Reviewer dsqW**
>
> Thank you for your precious comments. We have replied to your questions below and uploaded the revised paper with the revision shown in red.
>
> Q1: The theoretical results in Theorems 1 and 3 do not fully support the conclusion that underestimation leads to arbitrarily large aggregation and convergence errors, as they apply only to a specific robust aggregator, not all such aggregators. If a different robust aggregator is used, the effect of underestimation may not hold. This represents a notable limitation of the paper. Can the authors generalize the results in Theorems 1 and 3 to all robust aggregators to address this issue?
>
> A1: Not all $(\hat{f},\hat{\kappa})$-aggregators lead to arbitrarily large aggregation and convergence errors when there are actually $f$ Byzantine clients ($\hat{f}<f$). To show that, $(\hat{f},\hat{\kappa})$-robust aggregators include all $(f,\hat{\kappa})$-robust aggregators (by Theorem 2) that do not lead to arbitrarily large error.
>
> Our result well supports that "underestimation ($\hat{f}<f$) **can** lead to arbitrarily poor performance for both aggregators and federated learning" (in line 21), which is in contrast to non-underestimation case ($\hat{f}\ge f$) where all $(\hat{f},\hat{\kappa})$-robust aggregators lead to finite aggregation and guaranteed convergence.
>
> Q2: There is a discrepancy between the theoretical analysis and the empirical results, as the former is conducted in a deterministic setting, while the latter is based on a stochastic setting. This gap is significant, as stochastic noise can substantially affect the performance of FedRo and lead to different convergence properties. Can the authors extend the analysis to address this gap?
>
> A2: Good suggestion. We have extended to stochastic case in our appendix H in the revision, which indicates a trade-off result that is consistent with the deterministic case.
>
> Q3: In Table 1, the robustness coefficient $\kappa$ for the combination of $\hat{f}$-Krum and $\hat{f}$-NNM matches the lower bound in order. Why does the authors not use that aggregator in experiments? The reviewer would like to see the results of using $\hat{f}$-Krum $\circ$ $\hat{f}$-NNM in experiments.
>
> A3: Good suggestion. We have added $\hat{f}$-Krum $\circ$ $\hat{f}$-NNM to the additional experiments in Appendix G, which indicate results that are consistent with our previous experimental results.
>
> Q4: In Line 48, the authors claim that "Existing works typically require estimating the actual number $f$ or the fraction of the Byzantine clients to select the maximum number $\hat{f}$ that the aggregator can tolerate." To our knowledge, the tolerable maximum number of Byzantine clients (i.e., the breakdown point) for some robust aggregators is independent of the actual number $f$ or fraction of Byzantine clients, meaning that these terms do not need to be estimated in advance for such aggregators. For instance, $\frac{n}{2}$ for CWMed, $\frac{n-2}{2}$ for CWTM, and $\frac{n}{10}$ for centered clipping, with respect to the breakdown point. The authors should revise this statement to make it more precise and rigorous.
>
> A4: Good suggestion. We have changed to "To select an aggregator with a proper breakpoint $\hat{f}$ (namely, the number of maximum tolerable Byzantine clients) is essential to ensure good federated learning performance."
>
> Q5: In line 237, the authors claim that "since $\kappa$ of the commonly used aggregators like GM, CWTM, CWMed, and Krum do not match the lower bound even in the simple special case of $f = \hat{f}$." This statement is incorrect. As noted in Remark 2 of Allouah et al. (2023), CWTM does match the lower bound in order. The reviewer suggests that the authors carefully verify this statement.
>
> A5: We prove this statement correct below. First, Remark 2 of Allouah et al. (2023) said CWTM matches the lower bound in order when $f\le \frac{n}{2+\nu}$ for some constant $\nu>0$, but not for all feasible $f<\frac{n}{2}$. In fact, CWTM has $\kappa=\frac{6f}{n-2f}\big(1+\frac{f}{n-2f}\big)$ based on proposition 2 of Allouah et al. (2023), which has higher order than the lower bound $\frac{f}{n-2f}$ if $f=\lfloor \frac{n}{2}\rfloor-c$ for $c\ll n/2$. Second, Krum, GM and CWMed has $\kappa=6\big(1+\frac{f}{n-2f}\big)$, $\kappa=4\big(1+\frac{f}{n-2f}\big)^2$ and $\kappa=4\big(1+\frac{f}{n-2f}\big)^2$ respectively, based on Proposition 3-5 of Allouah et al. (2023) respectively. They have higher order than the lower bound $\frac{f}{n-2f}$ if $f\ll n$. We have added this explanation to the footnote in line 237. Thanks for pointing out.
>
> Q6: There are several typos in this paper. For instance, in line 225, $\frac{\hat{f}}{n - 2f}$ should be $\frac{\hat{f}}{n - 2\hat{f}}$, and in line 226, $\frac{\hat{f}}{n - 2\hat{f}}$ should be $\frac{\hat{f}}{n - \hat{f}}$. The authors are advised to carefully review the paper to address these and prevent any similar issues.
>
> A6: We have corrected these typos. Thanks for pointing out.

---

> > ### Comment · Reviewer_dsqW · 2025-11-24
> >
> > Thank you for the detailed explanations provided, which resolve most of my concerns. Therefore, I am increasing my score to 6. Good luck.

---

> > > ### Author Response · Authors · 2025-11-25
> > > **Thanks, Reviewer dsqW.**
> > >
> > > Thanks, Reviewer dsqW.
> > > Authors

---

### Official Review · Reviewer_RaGK · 2025-10-30

**Soundness:** 4
**Presentation:** 3
**Contribution:** 4
**Rating:** 6
**Confidence:** 5

**Summary:**

This paper provides a systematic theoretical analysis of the impact of estimating the number of Byzantine clients in Federated Learning (FL). The authors consider a scenario where an aggregator is chosen with a robustness parameter $\hat{f}$ (the estimated maximum number of Byzantine clients it can tolerate), while the true number is $f$. Their key contributions are:

Underestimation Risk: They rigorously prove that underestimation ($\hat{f} < f$) can lead to arbitrarily poor performance and divergence, even under favorable conditions like the Polyak-Łojasiewicz (PL) inequality.

Minimax Optimal Bounds for Non-Underestimation: For the non-underestimation case ($\hat{f} \geq f$), they establish matching lower and upper bounds (i.e., minimax optimal rates) for both the aggregation error and the convergence rate of the Federated Robust Averaging (FedRo) algorithm. These bounds are proportional to $\frac{\hat{f}}{n - f - \hat{f}}$.

Fundamental Trade-off: This bound reveals a fundamental trade-off: while an aggregator with a larger $\hat{f}$ can tolerate a wider range of attacks (any $f \leq \hat{f}$), its performance deteriorates when the actual number of attackers $f$ is small, as the error bound increases with $\hat{f}$.

Optimal Algorithm: They propose a novel composite aggregator ($\hat{f}$-Krum $\circ$ $\hat{f}$-NNM) that is proven to achieve the order-optimal upper bound without prior knowledge of the true $f$.

Empirical Validation: Experiments on CIFAR-10 validate the theoretical trade-off, showing performance collapse when $\hat{f} < f$ and performance degradation as $\hat{f}$ increases beyond $f$.

**Strengths:**

Novel Problem Formulation: Systematically studying the effect of the estimated number of Byzantine clients ($\hat{f}$) is a highly original and important direction.

Theoretical Completeness: The paper provides a complete minimax analysis, with tight lower and upper bounds for both aggregation error and algorithm convergence, under both underestimation ($\hat{f} < f$) and non-underestimation ($\hat{f} \geq f$) scenarios. The bounds, proportional to $\frac{\hat{f}}{n-f-\hat{f}}$, are rigorously derived.

Practical Relevance: The theoretical trade-off is clearly demonstrated and validated through experiments on a standard benchmark (CIFAR-10), connecting theory with practice.

Algorithmic Insight: The analysis of the composite aggregator ($\hat{f}$-Krum $\circ$ $\hat{f}$-NNM) provides a constructive method to achieve the order-optimal bound.

**Weaknesses:**

Computational Complexity of Optimal Aggregator: While the composite aggregator ($\hat{f}$-Krum $\circ$ $\hat{f}$-NNM) is theoretically optimal, it is computationally expensive. The per-round cost involves nearest neighbor searches for all clients, which may be prohibitive for very large-scale systems or high-dimensional models. The paper does not discuss efficient approximations or the practical scalability of this aggregator.

Limited Empirical Scope: The experiments, while supportive, are limited to one dataset (CIFAR-10), one model (ResNet-20), and one type of attack (Gaussian noise). Broader experimentation with more datasets (e.g., CIFAR-100, FEMNIST), model architectures, and sophisticated Byzantine attacks (e.g., label-flipping, backdoor) would strengthen the empirical validation.

Assumption of PL Condition: The convergence analysis for the upper bound (Theorem 5) relies on the Polyak-Lojasiewicz (PL) condition to achieve a global convergence rate. While the PL condition holds for some machine learning problems, it is a relatively strong assumption. Discussing the plausibility of this assumption in FL settings or exploring convergence under weaker conditions (e.g., just smoothness) would be valuable.

**Questions:**

The theoretically optimal aggregator ($\hat{f}$-Krum $\circ$ $\hat{f}$-NNM) appears computationally heavy for large $n$ and $d$, involving $O(n^2 d)$ operations per round. What are your thoughts on developing more computationally efficient aggregators (e.g., using approximate nearest neighbor search or sampling) that can still preserve, or nearly preserve, the theoretical guarantees? Would such approximations fit into your current theoretical framework?

Your experiments use a simple Gaussian noise attack. How do you expect your theoretical findings and the observed trade-off to hold under more complex and adaptive Byzantine attacks, such as those designed to mimic honest updates or perform model poisoning? Could such attacks potentially alter the established lower or upper bounds, for example, by affecting the heterogeneity constant $G^2$?

The convergence upper bound (Theorem 5) is derived under the PL condition, leading to an asymptotic error floor of $\mathcal{O}\big(\frac{\hat{f}G^2}{n-f-\hat{f}}\big)$. Can you provide any intuition or discussion on whether the core trade-off—that the asymptotic error floor scales with $\frac{\hat{f}}{n-f-\hat{f}}$—would still hold in non-PL settings, perhaps under different or weaker assumptions?

---

> ### Author Response · Authors · 2025-11-20
> **Authors' Response to Reviewer RaGK**
>
> Thank you for your precious comments. We have replied to your questions below and uploaded the revised paper with the revision shown in red.
>
> Q1: The theoretically optimal aggregator ($\hat{f}$-Krum $\circ$ $\hat{f}$-NNM) appears computationally heavy for large $n$ and $d$, involving $O(n^2 d)$ operations per round. What are your thoughts on developing more computationally efficient aggregators (e.g., using approximate nearest neighbor search or sampling) that can still preserve, or nearly preserve, the theoretical guarantees? Would such approximations fit into your current theoretical framework?
>
> A1: Interesting point. Consider the bucketing algorithm [1] with inputs {$x_i$}$_ {i=1}^n$, hyperparameter $s\in\mathbb{N}$ and output $y_i=\frac{1}{s}\sum _ {k=(i-1)s+1}^{\min(n,i\cdot s)}x _ {\pi(k)}$ for $i\in\{1,\cdots,\lceil n/s\rceil\}$ given random permutation $\pi$ on $[n]$, which requires fewer operations $\mathcal{O}(nd)$ per round. The random output from $\hat{f}$-Krum $\circ$ bucketing proposed by [1] has been proved to obtain the theoretically optimal bound but **in expectation**, so it perhaps fits into our current theoretical framework by adding $\mathbb{E}$ to our error metrics. $\hat{f}$-Krum $\circ$ $\hat{f}$-NNM proposed by [2] is the only aggregator to our knowledge that yields the optimal bound **deterministically**. (Omit authors in references below due to limited space)
>
> [1] Byzantine-robust learning on heterogeneous datasets via bucketing. ICLR 2022.
>
> [2] Youssef Allouah, Sadegh Farhadkhani, Rachid Guerraoui, Nirupam Gupta, Rafaël Pinot, and John Stephan. Fixing by mixing: A recipe for optimal byzantine ML under heterogeneity. AISTATS 2023.
>
> Q2: Your experiments use a simple Gaussian noise attack. How do you expect your theoretical findings and the observed trade-off to hold under more complex and adaptive Byzantine attacks, such as those designed to mimic honest updates or perform model poisoning? Could such attacks potentially alter the established lower or upper bounds, for example, by affecting the heterogeneity constant $G^2$?
>
> A2: Good question. In the revision, we have added experiments in Appendix G with label-flip and sign-flip attacks, which have the observed trade-off results that are consistent with our previous experimental results.  In the label-flip attack, a Byzantine client poisons its local dataset by replacing each ground-truth label with an incorrect label. In the sign-flip attack, a Byzantine client mimics honest updates but in the opposite direction.
>
> The theoretical results also still hold for different attacks. To elaborate, $G^2$ is a property of the objective function, which is not affected by attacks selected in the algorithm. Upper bounds apply for all attacks and thus are not affected by attacks. We select the provably optimal lower bounds obtained by a mild attack that exactly follows honest behavior, since other attacks may yield looser lower bounds.
>
> Q3: The convergence upper bound (Theorem 5) is derived under the PL condition, leading to an asymptotic error floor of $\mathcal{O}\big(\frac{\hat{f}G^2}{n-f-\hat{f}}\big)$. Can you provide any intuition or discussion on whether the core trade-off—that the asymptotic error floor scales with $\frac{\hat{f}}{n-f-\hat{f}}$—would still hold in non-PL settings, perhaps under different or weaker assumptions?
>
> A3: To our knowledge, PL condition is the weakest condition to ensure global convergence rate in federated learning works [1-4], and some federated learning works even rely on stronger assumption of strong convexity [5,6]. By intuition, PL condition that $\ell _ {\mathcal{H}}(w)-\ell ^ * \le\frac{1}{2\mu}||\nabla\ell _ {\mathcal{H}}(w)||^2$ can be almost hardly weaken to achieve the limiting error bound $\kappa G^2$ (as the number of iterations $T\to +\infty$) in our Theorem 5. To elaborate, consider a simple case where all the honest clients $k\in\mathcal{H}$ have the same objective function $\ell_k\equiv \ell _ {\mathcal{H}}(w)$, so $G=0$. Then for any stationary point $w$ such that $\nabla\ell _ {\mathcal{H}}(w)=0$, $w$ should be optimal to ensure our bound $\kappa G^2=0$. Otherwise, the algorithm may stuck at the non-optimal point $w$ which yields error bound$>0$. Therefore, it is natural to have the global optimality gap $\ell _ {\mathcal{H}}(w)-\ell^*$ dominated by the gradient norm $||\nabla\ell _ {\mathcal{H}}(w)||$, such as PL condition. (brief references below due to limited space)
>
> [1] Linear convergence of decentralized fedavg for PL objectives: The interpolation regime. 2025
>
> [2] FedAvg for Minimizing Polyak-Łojasiewicz Objectives: The Interpolation Regime. 2023.
>
> [3] A novel framework for the analysis and design of heterogeneous federated learning. 2021
>
> [4] On the convergence of local descent methods in federated learning. 2019
>
> [5] Federated learning’s blessing: Fedavg has linear speedup. 2021
>
> [6] On the Convergence of FedAvg on Non-IID Data. 2020

---

> > ### Comment · Reviewer_RaGK · 2025-11-24
> > **Thanks for your response**
> >
> > Thanks for your detailed explanation and careful response. In general, your response solves my concerns. I decide to maintain my score. Moreover, it would be better if you compare with the following recent works:
> >
> > Shiyuan Zuo et al. Federated learning resilient to byzantine attacks and data heterogeneity. IEEE Transactions on Mobile Computing, 2025.
> >
> > Puning Zhao et al. A huber loss minimization approach to byzantine robust federated learning. AAAI 2024.

---

> ### Author Response · Authors · 2025-11-27
> **Lit comparison for Reviewer RaGK**
>
> Thank Reviewer RaGK for bringing positive feedback and more papers.
>
> [1] proposes RAGA algorithm which is a special case of our FedRo algorithm using Geomed aggregator. [1] uses strong assumption of bounded gradient that we do not have and bounded individual heterogeneity $\theta _ m:=||\nabla \ell _ m(w)-\nabla \ell(w)||\le \theta$ for each client $m$, while we use weaker assumption of bounded average heterogeneity $\frac{1}{|\mathcal{H}|}\sum _ {m\in\mathcal{H}} \theta_m^2\le G^2$ among honest clients $m\in\mathcal{H}$. It can be proved that $G^2\le \theta^2$. For nonconvex analysis, [1] has gradient norm square converges to a gap proportional to $\theta^2$ with rate $T^{-(2/3-\delta)}$, while our algorithm converges to a smaller gap proportional to $G^2$ with faster rate $T^{-(2/3)}$.
>
> Similarly, [2] also studies a special case of our algorithm with their proposed Huber-loss-based aggregator and only 1 local update for each client. For nonconvex analysis, [2] also uses strong assumption of bounded gradient that we do not have and stronger heterogeneous assumption that $\sup _ {\|v\|=1} \mathbb{E}\Big[e^{\lambda v^T[\nabla\ell _ i(w)-\nabla \ell(w)]}\Big] \leq e^{\frac{1}{2} \sigma_\mu^2 \lambda^2}$, and it can be proved that $\sigma_{\mu}^2$ is larger than our $G^2$. The gap involves not only ${\rm constant}\times\sigma_{\mu}^2$ but also additional positive terms, while ours involves only ${\rm constant}\times G^2$ .
>
> [1] Shiyuan Zuo et al. Federated learning resilient to byzantine attacks and data heterogeneity. IEEE Transactions on Mobile Computing, 2025.
>
> [2] Puning Zhao et al. A huber loss minimization approach to byzantine robust federated learning. AAAI 2024.

---

### Meta-Review · Area_Chair_kGnv · 2026-01-07

**Summary:**

* Computational Scalability: Reviewers (RaGK, dsqW) noted that the theoretically "optimal" composite aggregator has a high computational complexity of $O(n^2d)$, making it potentially impractical for large-scale systems

* Novelty and Scope: Reviewer teAR argued that the "underestimation" result (catastrophic failure when $\hat{f} < f$) is simply a restatement of the well-known "breakdown point" in robust statistics. They also questioned if the new error bounds were significantly different from existing bounds.

* Empirical Validation: Initial concerns were raised regarding the limited scope of experiments, which originally used only one dataset (CIFAR-10) and one basic attack type (Gaussian noise).

* Theoretical Assumptions: The reliance on the PL condition for convergence was viewed as a relatively strong assumption for general federated learning.

**Reviewer Concerns:**

Concerns Addressed

* Empirical Scope: The authors addressed this by adding experiments in Appendix G featuring label-flip and sign-flip attacks, as well as testing the previously omitted composite aggregator

* Stochastic Settings: Authors extended their theoretical analysis to include the stochastic case (Appendix H)

* Technical Corrections: Typos in the bounds and inaccuracies regarding the optimality of existing aggregators (like CWTM) were corrected or clarified

* Clarification of Underestimation: The authors conceded that while the underestimation theory is complementary to existing concepts, it is necessary to complete the "trade-off" narrative they are proposing

Outstanding Concerns

* Computational Efficiency: While the authors discussed a "bucketing" algorithm as a more efficient alternative ($O(nd)$), they admitted it only achieves the optimal bound in expectation, whereas the proposed $O(n^2d)$ method is the only one known to do so deterministically

* Bound Significance: Reviewer teAR remained skeptical about whether the new bounds provide meaningful insights over previous ones in non-edge cases. The authors provided a final clarification on "order-level overestimation", but the interesting case happens when the estimation is close to the ground truth.

* Limited Significance: This work does not provide algorithms to estimate the number of Byzantine clients; rather, provide analysis on the effects of the estimation on convergence.

**Reviewer Scores:**

* RaGK: Explicitly stated they were maintaining their score of 6 after their concerns were solved.


* dsqW: Stated that the detailed explanations resolved most concerns.


* teAR: Remained the most critical about the significance of the analysis.

---

### Decision · Program_Chairs · 2026-01-26

Reject